# Roll-to-plate 0.1-second shear-rolling process at elevated temperature for highly aligned nanopatterns

Junghyun Cho[1], Jinwoo Oh[1], Joona Bang[2], Jai Hyun Koh [3], Hoon Yeub Jeong[1], Seungjun Chung [1,4] & Jeong Gon Son [1,4] ✉

The shear-rolling process is a promising directed self-assembly method that can produce high-quality sub−10 nm block copolymer line-space patterns cost-effectively and straightforwardly over a large area. This study presents a high temperature (280 °C) and rapid (~0.1 s) shear-rolling process that can achieve a high degree of orientation in a single process while effectively preventing film delamination, that can be applied to large-area continuous processes. By minimizing adhesion, normal forces, and ultimate shear strain of the polydimethylsiloxane pad, shearing was successfully performed without peeling up to 280 °C at which the chain mobility significantly increases. This method can be utilized for various high-χ block copolymers and surface neutralization processes. It enables the creation of block copolymer patterns with a half-pitch as small as 8 nm in a unidirectional way. Moreover, the 0.1-second rapid shear-rolling was successfully performed on long, 3-inch width polyimide flexible films to validate its potential for the roll-to-roll process.

Block copolymer (BCP) self-assembly can spontaneously form nanoscale structures ranging from 5 nm to 100 nm through microphase separation[1,2]. Therefore, directed self-assembly (DSA) of the BCP has been considered one of the alternative patterning methods of next-generation lithography[3,4]. One of the key challenges facing the semiconductor industry in recent years is further reducing device dimensions to 5 or 3 nm nodes through EUV lithography. However, there are still many issues that must be addressed, such as critical dimension (CD) uniformity and line-edge roughness, to develop narrower pattern processes such as sub-2 nm nodes. To address these issues, companies and research institutes are considering to combine DSA technology with EUV lithography[5,6]. Despite the possibilities, the simultaneous achievement of the perpendicular orientation of nanodomains for high-aspect-ratio[7], well-defined line-edges, long-range-order, and defect-free nanopatterns remains a challenge.

In order to overcome these challenges, different strategies for DSA have been developed, including graphoepitaxy[8–10], chemoepitaxy[11–16]

and shear alignment[17–24] methods. In particular, shear alignment has received attention as it enables the unidirectional alignment of anisotropic BCP microdomains in thin films, including cylinders and lamellae, to create clear line-space patterns. Initially, Register et al. reported a static shear method[18,25] for achieving unidirectionally aligned monolayer cylinders in BCP thin films during thermal annealing. The technique used a conformally contactable elastomeric polydimethylsiloxane (PDMS) pad as a buffer layer and weights for static shearing. Next, static shear and subsequent solvent annealing were attempted to eliminate defects and improve orientation order[24]. However, static shear should apply to lateral force proportional to the substrate area (challenging to scale up), and there is a risk of macroscopic delamination at high temperatures caused by significant displacement. Cold zone annealing with soft-shear (CZA-SS)[21,23] and laser-zone annealing[26–28] are sequential shear alignment methods by localized and directional thermal expansion of the PDMS pad using intense thermal gradients generated by highly localized heating. However, the

[1]Soft Hybrid Materials Research Center, Korea Institute of Science and Technology (KIST), Seongbuk-gu, Seoul 02792, Republic of Korea. [2]Department of Chemical and Biological Engineering, Korea University, Seongbuk-gu, Seoul 02841, Republic of Korea. [3]Clean Energy Research Center, Korea Institute of Science and Technology (KIST), Seongbuk-gu, Seoul 02792, Republic of Korea. [4]KU-KIST Graduate School of Converging Science and Technology, Korea University, Seongbuk-gu, Seoul 02841, Republic of Korea. ✉e-mail: jgson@kist.re.kr

CZA-SS method has difficulty controlling the expansion/contraction direction of the PDMS located on the top. On the other hand, the laser-shear method requires an additional heat absorption layer or base temperature controls[29] on the substrate to provide appropriate mobility to BCP chains. A shear-rolling process[17] has been recently developed, which can apply a large amount of shear stress to the BCP film by varying the speed of the roller and substrate, realizing a 4-inch scale unidirectional orientation and stretchable chiral metamaterials[22]. Nonetheless, the process was carried out at low temperatures at 150 °C to avoid massive peeling, resulting in the need for repeated rolling for high-quality orientation. However, even in DSA on the chemical pattern (chemoepitaxy), it was possible to realize more ordered patterns in a few minutes at a high temperature of over 250 °C[30], compared to the conventional annealing required for several hours or more at a low temperature of less than 200 °C[11,12]. Therefore, the shear-rolling process also should be performed at a much higher temperature to realize higher quality patterns and enable continuous roll-to-roll processes.

In this paper, we introduce a 0.1-second rapid shear-rolling process at very high temperatures to achieve a highly ordered and unidirectionally aligned perpendicular lamellar structure over a large area with a single rolling. It is crucial to avoid film instability to ensure a high-quality alignment pattern in a short time at a high temperature. By reducing the adhesion and normal stress to minimize friction with the PDMS and increasing the processing speed to minimize the ultimate shear strain and contact time of the PDMS, the shear-induced highly ordered unidirectional alignment with fewer defects was successfully achieved without macroscopic delamination up to 280 °C maximizing the chain mobility of the BCP. Additionally, polystyrene-*block*-poly (methyl methacrylate) (PS-*b*-PMMA), which originally forms a parallel orientation of microdomains on top due to contact with the PDMS, can also be perpendicularly oriented and unidirectionally aligned without the need for special surface treatment due to the very short shear/contact time. Furthermore, using the different neutralization strategies such as filtered plasma[7] and polarity-switching top coat[31], perpendicularly oriented and straightly aligned line-space patterns were created through 0.1-second shear-rolling. This was achieved with a high-χ triblock BCP and a silicon-containing high-χ BCP with half-pitches of 10.5 nm and 8 nm, respectively. Finally, we successfully demonstrated the shear-rolling process on a flexible polyimide

substrate, resulting in unidirectional and perpendicular alignment of the BCP films over a large area, confirming its potential application in roll-to-roll processes.

## Results

### Bumpy approaches for 0.1-second shear-rolling process at high temperatures

Figure 1 shows a schematic illustration of the high temperature and 0.1-second rapid shear-rolling process to realize high-quality nanopatterns of unidirectionally aligned perpendicularly oriented BCP lamellar microdomains. The BCP films on the Si wafer are placed on a temperature-controllable hot stage, with a PDMS buffer pad between the roller and the film. Various parameters must be optimized to successfully perform the ultra-high temperature and ultra-fast shear-rolling process. If polymer chains with high mobility at ultra-high temperatures are rapidly arranged under a uniform shear force, unidirectional orientation with a high degree of alignment can be achieved with a single shear process. However, the high mobility of the polymer chains can also lead to large-scale polymer movements, such as massive film peeling and surface undulation. In many BCP shear studies reported, including our previous studies[17], the shear temperature was executed up to only 170 °C[24,32,33], mainly due to the high flow ability of the BCP at higher temperatures that caused friction-dominant large-scale film damages. To avoid these phenomena, adhesion properties between the PDMS and the film should be reduced to minimize the transfer of unwanted forces, from PDMS deformation during shear and subsequent recovery after shear. It is also necessary to minimize interfacial instability by reducing the normal force during shear, and to minimize the ultimate shear strain when shearing is completed to enable only microscopic orientational movement of the polymer chain and prevent unnecessary large-scale movement.

To optimize the parameters related to shear-rolling, 61 nm thick (~2 $L_0$) PS-*b*-PMMA (45 kg mol⁻¹, 23k – 22k, lamellar morphology) films were primarily used. A hydroxyl-terminated random copolymer brush, PS-*r*-PMMA-OH, was grafted on a Si wafer to neutralize the bottom interface. In order to prevent large-scale damage to the film due to rising temperatures, we first controlled the surface of the PDMS pads in contact with the BCP films. The interfacial properties between the

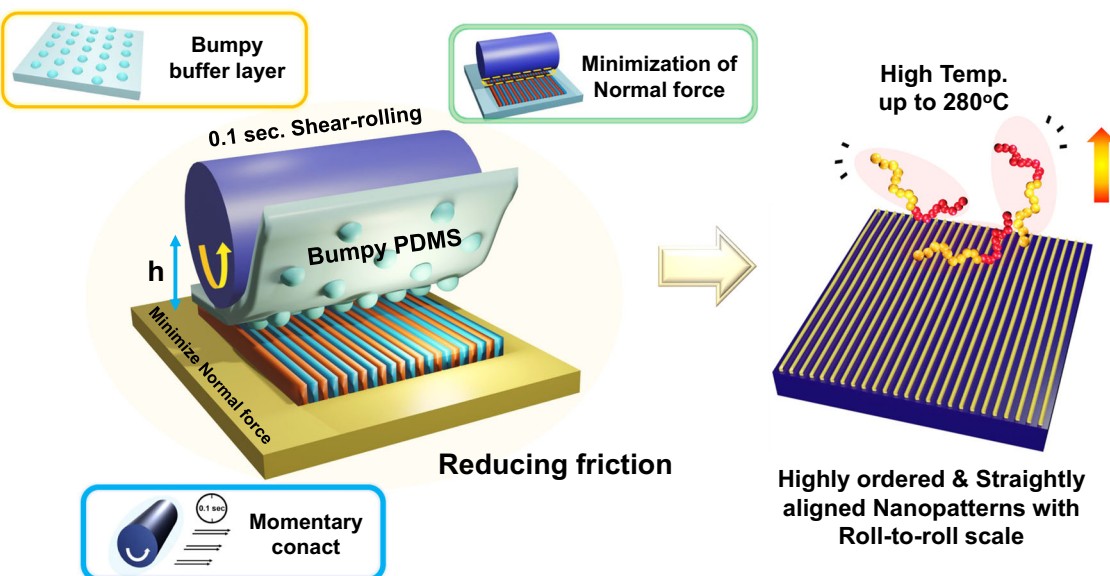

**Fig. 1 | Schematic illustrations of the roll-to-plate 0.1-second shear-rolling process at high temperatures of 280 °C.** Using bumpy elastomer pads, reducing the normal force, and minimizing the maximum shear strain (contact time) during

shear-rolling engage only monotonic unidirectional shears without any massive delaminations or undulations.

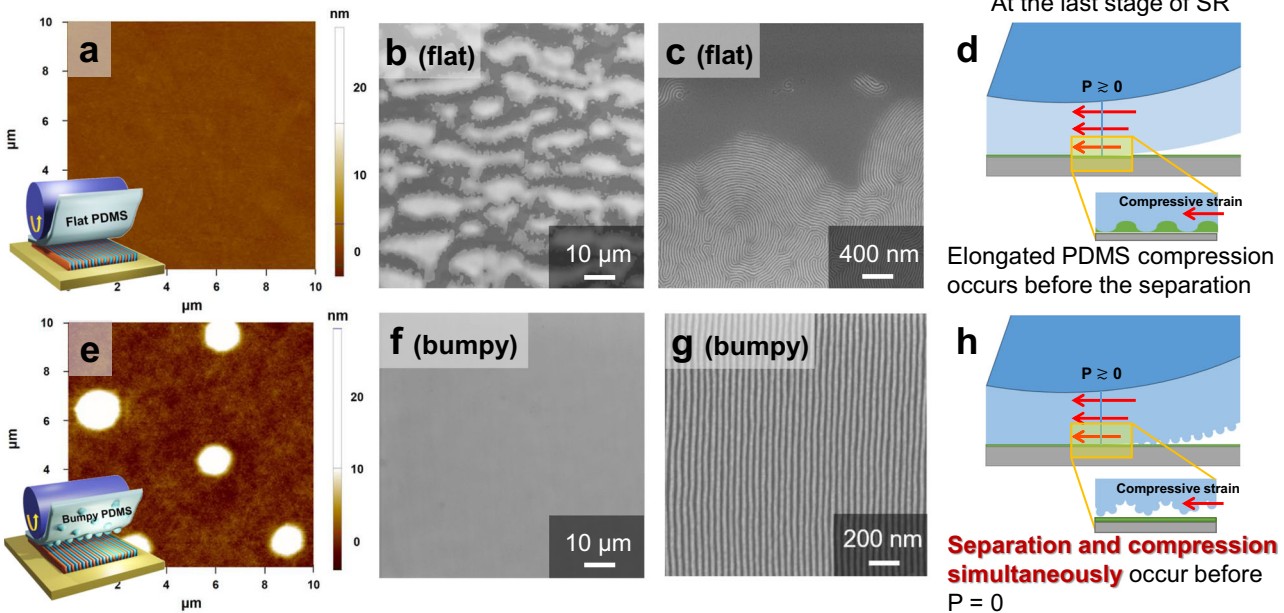

**Fig. 2 | Topographical difference between flat and nano-bumpy PDMS pads.**
**a**, **e** AFM surface morphology images of (**a**) flat PDMS ($R_q = 0.686$) and (**e**) bumpy
PDMS pads. **b**, **f** SEM low-magnification topographical images and **c**, **g** their high
magnification images for nanodomain orientations of shear-rolled PS-*b*-PMMA

films at 280 °C with (**b**, **c**) flat PDMS pad and (**f**, **g**) bumpy PDMS pads.
**d**, **h** Schematics of the shear-rolling process at the last stage with (**d**) flat PDMS and
(**h**) bumpy PDMS.

PDMS pad and BCP film can significantly influence the surface
instability of the film under shear, leading to a decline in the orienta-
tional order of BCP microdomains. It is generally believed that a very
flat PDMS pad should be used to apply a uniform shear force to the BCP
film on an atomically flat Si wafer. Therefore, the PDMS pad for the
shear experiment is usually cured at 60 °C, 12 h using a Si wafer as a
mold, as shown in Fig. 2a with an RMS roughness of ~0.7 nm, which is
similar to our previous paper. However, in contrast to the above, we
have prepared a PDMS pad with nano-bumps with an average size of
~1.5 µm and a height of ~22 nm by protruding an uncured PDMS pre-
polymer at a higher temperature through two-step curing as shown in
Fig. 2d. This is a well-known process where a specific part of a spherical
particle is extruded using two-step curing to synthesize a snowman-
shaped polymer particle[34].

The PDMS pads with different surface morphologies significantly
impacted the BCP surface uniformity, after undergoing shear-rolling at
roller/substrate speeds of 36/32 mm s$^{-1}$ at 280 °C. Figure 2b shows a
low-magnification SEM image of shear-rolled PS-*b*-PMMA film with a
flat PDMS pad, showing elongated undulations orthogonal to the shear
direction spanning the entire area with a period of several tens of
micrometers. Because of the flow that formed these unwanted mac-
roscopic undulations, the BCP microdomains were not uni-
directionally oriented, as shown in Fig. 2c. Additionally, at larger
ultimate shear strain conditions, such as 3.6 (10/6 mm s$^{-1}$), very severe
irregular delamination and slipping in stick-slip mode[35] occurs, as
shown in Supplementary Fig. S1. On the other hand, the bumpy PDMS
pad maintained a flat BCP film without any macroscopic structures,
even after being subjected to shear-rolling at 280 °C (Fig. 2f). The
surface of the BCP film did not show any stamping traces caused by the
bumpy PDMS structure (medium magnification of SEM image in Sup-
plementary Fig. S2 and AFM topographical image in Supplementary
Fig. S3). In addition, by modeling the distribution of the stress field
during intermittent contact in Supplementary Figs. S4 and S5 in sup-
porting information, simulation results confirm that a uniform shear
stress can be applied when such contact occurs. Because only shear
was applied effectively to the flat BCP films without macroscopic
undulations or flows, the unidirectional orientation of BCP

perpendicular lamellar nanostructures was successfully achieved, as
shown in Fig. 2g.

To investigate why flat PDMS forms undulations and bumpy
PDMS forms flat BCP films, we focused on the last step of shear-rolling,
where the ultimate shear strain is the highest (~0.4 for normal shear
conditions in Supplementary Fig. S6), as shown in Fig. 2d, h, and
Supplementary Fig. S7. Just before the detachment between BCP film
and PDMS, due to the elastic properties of PDMS, which has already
undergone a lot of elongated deformation, the returning force
becomes stronger according to Hooke's law. Compressive stress is also
transferred to the still attached BCP film due to the compressive strain
of the elongated PDMS. Here, we think that instabilities in compressed
film-substrate systems, which explain the forming of regular period
wrinkles when a thin film on a stretchable substrate is subjected to
compressive stress[36], can be applied. Before the PDMS and BCP films
are separated, that is, just before the pressure becomes 0, wrinkles can
form in the BCP film with different modulus due to the compressive
deformation of PDMS. Here, for flat PDMS, detachment requires a
small but additional force (the force from the roller lifting the PDMS)
after the pressure becomes 0 due to adhesion (Work of adhesion
between PDMS and PS/PMMA ~ 34.5 mN/m at 200 °C)[37]. Therefore,
because there is sufficient time for the compressive stress of PDMS to
be transferred to the BCP film before detachment (Fig. 2d), relatively
regular undulations can be formed in the BCP film. On the other hand,
for bumpy PDMS, due to the pointed structural nature and elasticity of
the bumps, the contact area decreases rapidly and separation also
occurs before the pressure reaches 0. At this time, compression of the
elongated PDMS and separation can occur simultaneously before the
pressure becomes 0 (Fig. 2h), thereby avoiding the formation of
wrinkles due to compressive strain. Thus, even at a high temperature
of 280 °C, only shear can be effectively transmitted without uneven
surface instability, resulting in well-aligned BCP line-space nano-pat-
terns throughout the entire area.

### Pressing heights effects on 0.1-second shear-rolling process
As the next strategy to prevent large-scale damage to the film, the
normal force pressing down the film was controlled by adjusting the

relative height of the roller from the BCP surface. During static shear, a high lateral force is applied to the film, causing slippage of the PDMS pad when it exceeds the static frictional force. Therefore, to increase the static friction force, the magnitude of the normal force must increase, leading to unwanted PDMS deformation and instability at the pad-film interface. However, in the shear-rolling process, the shear force is only applied in a narrow contact area, allowing for effective shear without slipping even with low normal force. The height at which the roller and the pad start to contact was set to 0, and rapid shear-rolling with a shear rate of $5.0\,s^{-1}$ was performed while increasing the pressed height from −0.02 to −0.06, −0.10, and −0.14 mm. Through the radius of the roller and the height between the roller and pad, the increase in contact width due to deformation could be calculated as 2.2, 3.8, 4.9, and 5.8 mm, respectively, and it was experimentally confirmed that the contact widths were formed at similar values through ink stamping (Supplementary Fig. S8).

When shear-rolling was performed at 280 °C, $36/32\,mm\,s^{-1}$ of roller/substrate speeds, and $\sim5\,s^{-1}$ of shear rate, there were almost no macro-scale topographic features at −0.02 mm height (ultimate shear strain (USS) ~ 0.34, Fig. 3a). However, at heights of −0.06 (USS ~ 0.59) and −0.10 mm (USS ~ 0.77), elongated undulation patterns began to appear orthogonal to the shear direction with periods of ~10 μm (Fig. 3b). The non-uniformity became more pronounced at a higher pressure height of −0.14 mm (USS ~ 0.90, Fig. 3c). This instability phenomenon often appears in the soft nanoimprinting process with PDMS, which creates nanoscale patterns on thin films by applying pressure (normal force), in the form of topographical undulation in the area where high pressure is concentrated[38]. Increasing the pressing height during rolling causes additional lateral tensile strain by compressing the entire PDMS pad. This elongation of the pads becomes an additional source for applying compressive strain when returning after shear. Furthermore, as the pressing height increases, the contact width increases, leading to longer shear application time. Consequently, the ultimate shear strain (USS) increases, resulting in a larger compressive

strain after shearing. These large compressive strains after shearing eventually lead to film undulation (Supplementary Fig. S9). Here, the results, where undulation becomes more pronounced as USS increases and there is no significant difference in the period, are consistent with the compressive strain-dependent wrinkling phenomena of thin films on elastomers.

In addition, in Fig. 3e, the macroscopic instability of the film surface for various normal forces and shear rates was investigated and marked with different square symbols. As the normal force and shear rate increase, the ultimate shear strain of PDMS increases, resulting in severe undulation of the film. In Fig. 3d, at a high shear rate of $10\,s^{-1}$ with higher pressing heights of −0.10 (USS ~ 1.53) and −0.14 mm (USS ~ 1.80), more prominent ridges of several tens of micrometers parallel to the shear direction start to appear. These ridges are thought to be caused by applying shear while PDMS wrinkles are formed in the deformation direction due to the high Poisson's ratio of PDMS (~0.5) when extreme shear deformation occurs in the PDMS pad with low adhesion. Along with these microstructures, the orientations of BCP nanopatterns were also observed under the same conditions (inset images of Fig. 3a–d) and visualized as a contour plot with 2D Herman's order parameters[39–42] in Fig. 3e (See the Supplementary information and Supplementary Figs. S10 and S11 for the detailed information about Herman's orientation parameter). Under the given conditions, a high orientation order parameter of 0.98 or more was shown when the film's macroscopic structure was almost absent. A relatively high degree of orientation was maintained even with slight undulations. However, when macroscopic undulations of certain sizes or more are formed, non-negligible flow in a direction different from the shear occurs, and the flow significantly reduces the degree of orientation of the nanopattern (Supplementary Fig. S12). Based on these results, it was found that shear must be applied under conditions that minimize the formation of unstable structures to achieve a highly oriented BCP film.

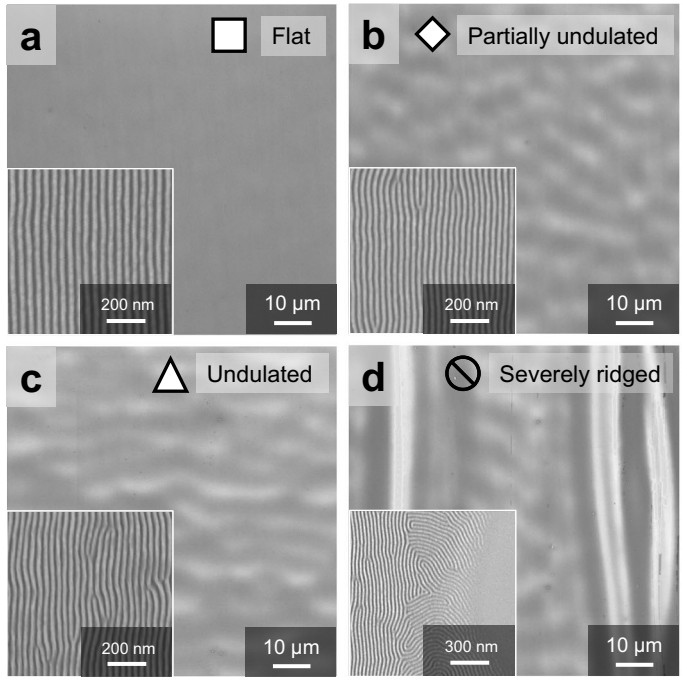

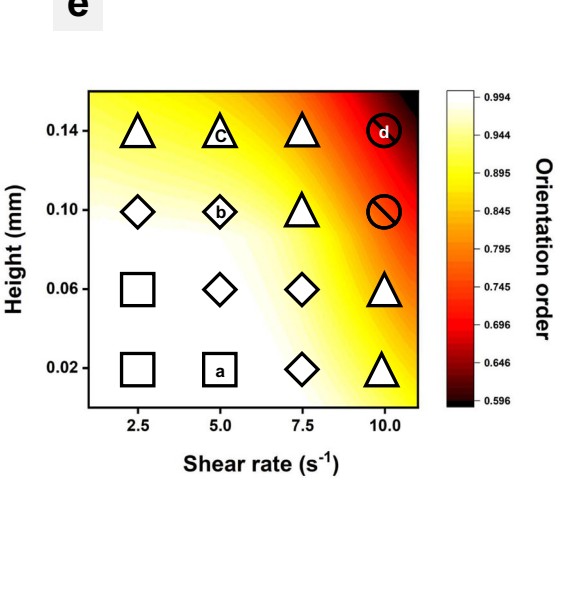

**Fig. 3 | Macroscopic and nanoscopic morphology changes depending on normal forces and shear rates.** SEM images of the macroscopic morphologies and nanostructure orientations (inset) of the PS-*b*-PMMA films as the pressing heights were varied with (**a**) h = −0.02 mm, (**b**) h = −0.10 mm, and (**c**) h = −0.14 mm during shear-rolling at a shear rate of $5.0\,s^{-1}$. **d** SEM images of macroscopic instability and nanostructure orientation (inset) of the PS-*b*-PMMA films at a pressing height of −0.14 mm and a shear rate of $10\,s^{-1}$. **e** The macroscopic instability is indicated by different symbols and a color-coded contour plot of nanoscopic orientation order parameters against pressing height and shear rate. Source data are provided as a Source Data file.

## Contact time effects on 0.1-second shear-rolling process

The third strategy is to speed up the shear-rolling process to reduce the contact time and limit the maximum shear strains of the PDMS, as shown in Fig. 4. The substrate speed ($v_x$) varied from 8 to 20 mm s$^{-1}$, while maintaining a constant shear rate of 2.5 s$^{-1}$. Since the height between the roller and the film is set at −0.02 mm, the contact width is only 2.2 mm, resulting in an actual shear process time of only 0.1 to 0.28 s and USS from 0.28 to 0.69. When the speed is set to a low value of 8 mm s$^{-1}$, as shown in Fig. 4a, the same shear rate is applied for a longer duration, leading to larger ultimate shear strain (USS ~ 0.69) and the formation of macroscopic structures on the BCP film. The grids observed were a result of both the longitudinal ridges caused by the vertical contraction of the shear-deformed high Poisson's ratio PDMS pad during shear (as explained in Fig. 3d) and the parallel undulations from the PDMS returning after the shear. These grids contained longitudinal ridges that were formed independently and parallel undulations that were affected by them, allowing for the determination of their formation order. Furthermore, if the contact time between the PDMS and PS-*b*-PMMA film is too long, there can be preferential wetting between the PDMS and the PS domain with low interfacial energy, resulting in the parallel orientations of BCP lamellae (inset of Fig. 4a) and consequent formation of terrace morphologies in Fig. 4a[20]. As the contact time with PDMS decreases, the perpendicular orientation of PS-*b*-PMMA appears to become dominant. Still, even at a $v_x$ of 12 mm s$^{-1}$ (USS ~ 0.46) and 16 mm s$^{-1}$ (USS ~ 0.34), micro-ridges and undulations can have a negative effect on the film's orientation order (Fig. 4b, c). At speeds of 20 mm s$^{-1}$ (USS ~ 0.28) and above, high-quality vertical line-space patterns were achieved with negligible macro-morphological formation in Fig. 4d. The rapid shear-rolling can minimize the unwanted movement caused by the deformation of the PDMS pad by only applying shear for a very short time. In addition, this rapid shear-rolling successfully achieved a perpendicular orientation of PS-*b*-PMMA, which have neutral conditions on the top surface with air at temperatures above 200 °C (Supplementary Fig. S13)[30,43], without any surface treatment due to a short contact time with the PDMS pad of the order of 0.1 s, as also can be seen in Supplementary Fig. S14. Here again, as the speed of the substrate increases, the undulation diminishes and there is no significant difference in the period, which can be judged to be caused by the wrinkling phenomenon from compressive deformation rather than the large velocity-originated slip-stick phenomenon.

## Orientations of shear-rolled BCPs at different temperatures

Figure 5 displays the results of our investigation into the orientations of shear-rolled PS-*b*-PMMA films at different temperatures. Three conditions were optimized – pressing height of −0.02 mm, contact time of 0.07 s, the use of a nano-bumpy pad – to minimize instability. SEM images, orientational color image analysis and Herman's order

parameters were utilized for these results. At a relatively low temperature of 220 °C, most areas were not directionally aligned, and fingerprint-like structures with low order parameters were formed (Fig. 5a). This indicates that a one-step, rapid shear process is not enough to align the slow polymer chains at low temperatures. When the temperature is raised to 240 and 260 °C (Fig. 5b, c), the BCP chains react to shear and become highly aligned in the shear direction. This alignment results in increased order parameters to over 0.98 and low line edge roughness (LER) of 3.5 nm (Supplementary Fig. S15), although some defects may still be present. By further raising the temperature to 280 °C, the mobility of the polymer chains was increased, resulting in highly ordered and straight line-space BCP patterns (Fig. 5d). These patterns showed an orientation order of 0.993 which is very close to the perfect orientation of 1 and had few defects such as dislocations and bridges (Fig. 5f) that amounted to less than 1 per µm$^2$. Also, the LER did not increase significantly to 3.7 nm. As temperature rises, the mobility of BCP film increases, and as a result, it is evident that the orientation orders continuously enhance up to 280 °C (Fig. 5g). However, at a very high temperature of 290 °C, it was observed that the random orientation reappeared. This could be a result of polymer degradation or a transition to disordered states (Fig. 5e). 2D patterns of grazing incident small-angle X-ray scattering (GISAXS) at 280 °C in Fig. 5h also showed highly ordered peaks up to the 7th fold in the lateral axis that matched the domain spacing of ~30.4 nm. In addition, the large-area SEM image also shows straightly aligned line space patterns (Fig. 5i) without defects over a large area. These confirm that the pattern alignment of rapid shear-rolling is perfect, covering a large area overall.

## The universality of rapid shear-rolling to high χ-BCPs

Beyond the PS-*b*-PMMA application that creates a neutral air interface, the 0.1-second rapid shear-rolling is also compatible with multiple neutralization strategies on top surfaces. This includes filtered plasma[7] and polarity-switching top coats[31], which are suitable for high-χ BCPs with significant surface energy differences, as shown in Fig. 6. Two high-χ BCPs, poly(2-vinylpyridine)-*block*-polystyrene-*block*-poly(2-vinylpyridine) (P2VP-*b*-PS-*b*-P2VP, $M_n$ ~ 12−24−12 kg mol$^{-1}$) and poly(4-trimethylsilylstyrene)-*block*-poly(4-methoxystyrene) (PTMSS-*b*-PMOST, $M_n$ ~ 17 kg mol$^{-1}$), were used for the creation of ultrafine nanopatterns with half-pitches of 10.5 nm and 8 nm, respectively. For P2VP-*b*-PS-*b*-P2VP, shear-rolling was performed with the filtered plasma strategy that we developed in previous studies[7], which creates perpendicular orientation by forming low-density cross-linking on the top surface of the film. We confirmed that filtered plasma-treated P2VP-*b*-PS-*b*-P2VP films can stably form perpendicular orientation at temperatures up to 245 °C (Supplementary Fig. S16a). It was also evident that the 0.1-second rapid shear-rolling can realize highly ordered line-space patterns with a 10.5 nm half-pitch at 230 °C. (Fig. 6a). Here,

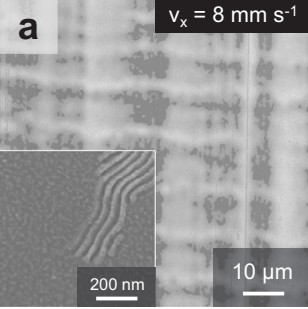
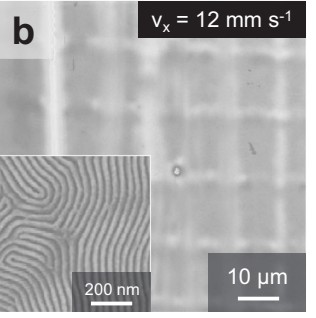
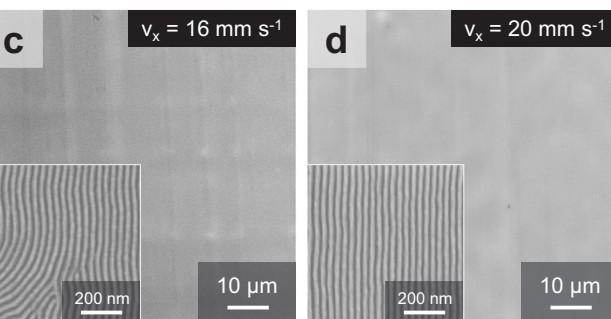

**Fig. 4 | Macroscopic morphologies and nanostructure orientations of different substrate speeds (contact time).** SEM images of the macroscopic morphologies and nanostructure orientations (inset) of the PS-*b*-PMMA films as the substrate speeds ($v_x$) were varied with (**a**) $v_x$ = 8 mm s$^{-1}$, (**b**) $v_x$ = 12 mm s$^{-1}$, (**c**) $v_x$ = 16 mm s$^{-1}$ and (**d**) $v_x$ = 20 mm s$^{-1}$ during shear-rolling at a shear rate of 2.5 s$^{-1}$.

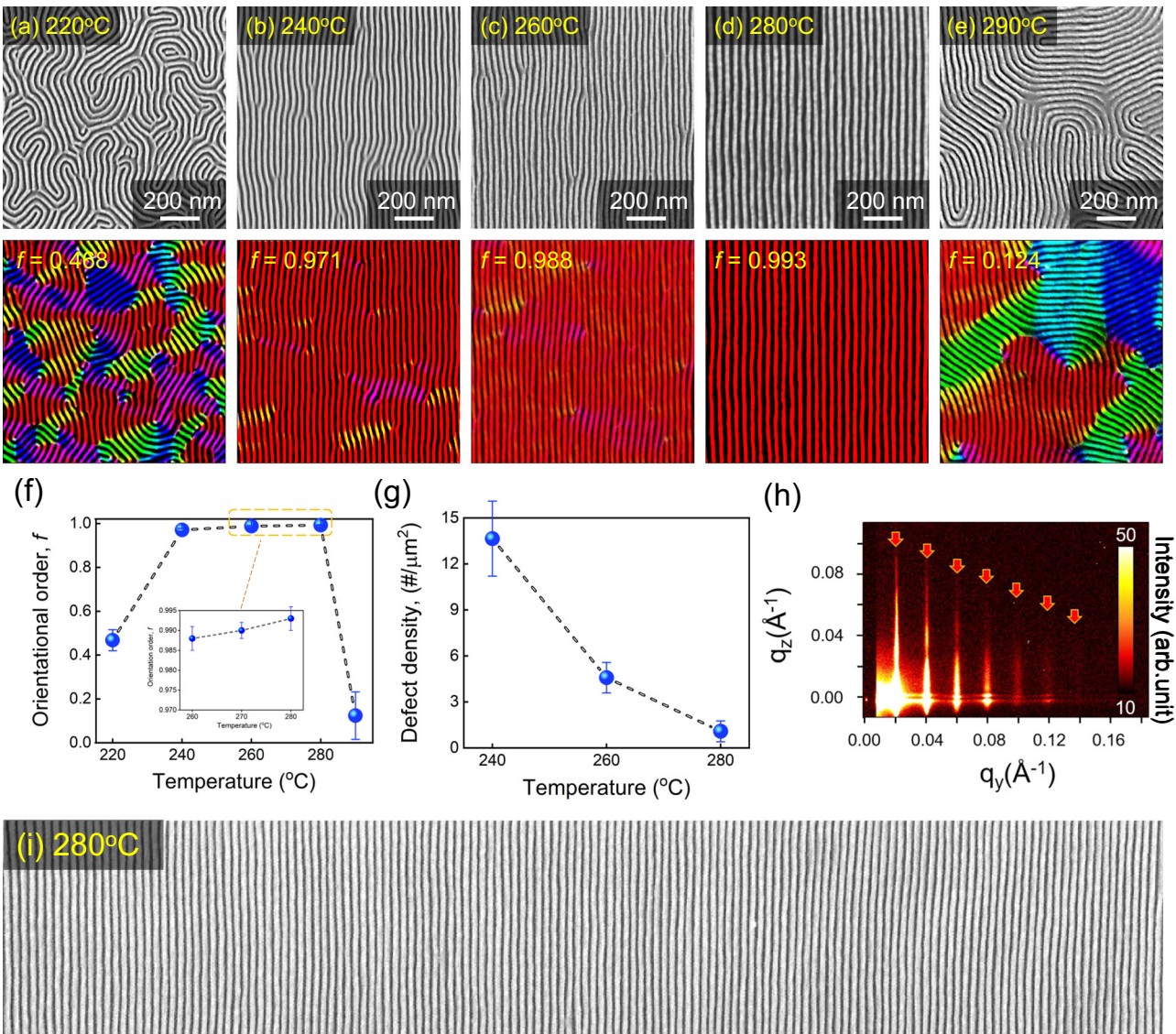

**Fig. 5 | Orientational analysis of the rapid shear-rolling process.** Nanostructural SEM images and orientational analysis of the rapid shear-rolling process at 36/32 mm s⁻¹ of roller/substrate speeds and optimized conditions with temperature changes. SEM images and their orientational order maps of shear-rolled PS-*b*-PMMA films at (**a**) 220, (**b**) 240, (**c**) 260, (**d**) 280, and (**e**) 290 °C. **f** The plot of the orientational order parameters and **g** defect (dislocations and bridges) densities of the BCP patterns as temperature changes. Error bar represent the standard error of measured orientational order from at least five independent experiments. **h** GISAXS 2D patterns and **i** large area SEM images of shear-rolled PS-*b*-PMMA films at 280 °C. Source data are provided as a Source Data file.

we observe the effect of lateral thermal gradients like soft-shear laser zone annealing[26–28] on our 0.1-second rapid shear rolling. After placing 0.7 mm thick PDMS on a BCP film and maintaining thermal equilibrium on a hot plate for 5 min to weaken the effect of the lateral thermal gradient, shearing was performed for 0.1 s (Supplementary Fig. S17a). Even after this, unidirectional orientation with a high degree of order was achieved in Supplementary Fig. S17b, allowing us to conclude that lateral thermal gradients do not significantly affect the quality of orientation. However, in original experiments where PDMS was attached to a roller, we believe that the lateral thermal gradient may have some effect on alignment.

To achieve a perpendicular orientation of the silicon-containing BCP (PTMSS-*b*-PMOST), a hydroxyl-terminated neutral brush (poly(4-*tert*-butylstyrene-*co*-3,4-methylenedioxystyrene), P(tBS-*co*-MDOS)−OH) was applied to the bottom interface and a polarity-switching top coat (poly((maleic anhydride-*alt*-3,5-di-*tert*-butylstyr-ene)-*co*-(maleic anhydride-*alt*-4-*tert*-butylstyrene))) was applied to the top interface. The compositions of both neutral brush and top coat

were carefully determined to neutralize the interfaces as described in the previous study[16]. These materials effectively neutralized both interfaces and allowed for the desired orientation of the BCP thin film. The structure of both the top coat and the perpendicularly oriented BCP thin films remained intact even at temperatures as high as 245 °C (Supplementary Fig. S16b). When shear-rolling was performed at temperatures exceeding 245 °C, the film surface exhibited instability, as shown in Supplementary Fig. S18. To prevent instability in the top coat and BCP films, we conducted shear-rolling at 235 °C, which pro-duced highly aligned and perpendicularly oriented patterns of 8 nm half-pitch PTMSS-*b*-PMOST (Fig. 6b). Here, it was found that highly aligned P2VP-*b*-PS-*b*-P2VP and PTMSS-*b*-PMOST films can be made at lower temperatures than the PS-*b*-PMMA. This indicates that the 0.1-second rapid shear-rolling is effective at a temperature range where the BCPs do not exceed the order-disorder temperature or decompose and top neutrality remains unchanged. We have confirmed that our shear-rolling process, lasting only 0.1 s, can be used for different neutralization strategies and BCP pairs.

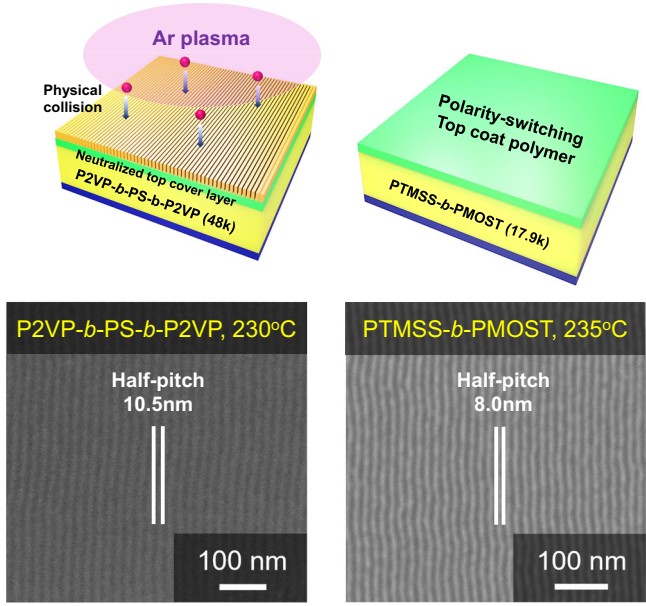

**Fig. 6 | 0.1-second rapid shear-rolling process with various neutral strategies and BCP pairs.** The straightly aligned and perpendicularly oriented (**a**) 10.5 nm half-pitch P2VP-*b*-PS-*b*-P2VP thin films with filtered plasma treated top surface at 230 °C and (**b**) 8 nm half-pitch silicon-containing, high-χ PTMSS-*b*-PMOST films with polarity-switching top coats at 235 °C.

## Roll-to-roll applicability of rapid shear-rolling

Our rapid shear-rolling process has the advantage of fabricating a large area at high speed through a continuous process in under 0.1 s of contact time. To take advantage of these features, the PS-*b*-PMMA film was coated on a 3-inch wide and 4-inch long flexible substrate suitable for a roll-to-roll process[44,45], and the 0.1-second rapid shear-rolling was executed subsequently. Figure 7a, b shows a photographic image and schematic illustration of the roll-to-plate rapid shear-rolling process on a flexible polyimide substrate. To ensure a reliable perpendicular orientation of PS-*b*-PMMA microdomains, the polyimide substrate was irradiated with UV-ozone for 30 min. Next, the PS-*r*-PMMA-OH random copolymer brush was coated and thermally treated at 180 °C for 24 h to neutralize the substrate. We confirmed that the neutralized polyimide substrate successfully formed a perpendicular orientation of PS-*b*-PMMA lamellae in thin films (Supplementary Fig. S19). On this PS-*b*-PMMA spin-coated 3-inch-wide and long flexible polyimide substrate, the 0.1-second rapid shear-rolling was executed at 36/32 mm s⁻¹ at 280 °C, as shown in Fig. 7b. The thin film orientations at nine regions departed by 32 mm vertically and 26 mm horizontally from the center of the flexible polyimide films (Fig. 7c and Supplementary Fig. S20) were investigated through SEM and their orientational order parameters as shown in Fig. 7d, e. In almost all areas of the large-scale polyimide film, the PS-*b*-PMMA line-space patterns were highly ordered and straight with orientation order values above 0.97, excluding region 3. These minor defects can associate with the lab-scale process and variations in the thickness of large-area PDMS at the edges. We finally achieve a 10 nm-level line-space pattern in a straight alignment directly on a 4-inch scale large, flexible substrate.

## Discussion

In this study, we demonstrated optimized conditions for ultra-high temperature and ultra-fast (less than 0.1 s) processing of the shear-rolling method. Introducing nano-bumpy structures to PDMS to reduce interfacial adhesion, lowering the pressing height to minimize normal force, and rapid process to minimize the ultimate shear strain of PDMS, thereby the movement of the polymer film due to the forces

other than shear could be minimized. As a result, our optimized shear-rolling can be processed at ultra-high temperatures of up to 280 °C, maximizing the mobility of BCPs and greatly improving the unidirectional alignment of BCP microdomains in thin films in a single run. Additionally, we have successfully achieved straightly aligned nano-patterns with perpendicular orientation from any BCP types including those with high-χ and sub−10 nm phase separation and through various neutralization methods. We also utilized our single-step patterning method to fabricate unidirectional nanopatterns on a 3-inch wide, long and flexible polyimide substrate, to apply the roll-to-plate process. This rapid, high temperature shear-rolling method allows BCP nanopatterns to be well aligned even in a continuous process such as roll-to-roll manufacturing. Therefore, this technique can be applied as a platform to fabricate high-quality nanostructures in a large area, contributing significantly to the large-area and commercialization of next-generation nano-devices such as waveguides for EUV that require large-area unidirectional nanopatterns of 6 inches or more or flexible nano-chiral metamaterials.

## Methods

### Materials and sample preparation

Poly(styrene-*b*-methyl methacrylate), PS-*b*-PMMA (23−22 kg mol⁻¹), hydroxyl-terminated poly(styrene-*r*-methyl methacrylate) random copolymer brush, P(S-*r*-MMA)−OH (14.2 kg mol⁻¹, $f_{PS} \approx 0.57$), poly(2-vinylpyridine)-*b*-polystyrene-*b*-poly(2-vinylpyridine), P2VP-*b*-PS-*b*-P2VP (12−24−12 kg mol⁻¹) and hydroxyl-terminated P(S-*r*-2VP)−OH brush (4.5 kg mol⁻¹, $f_{PS} \approx 0.7$) were purchased from PolymerSource Inc. A hydroxyl-terminated brush in toluene solution (1 wt %) was spin-coated onto ultraviolet ozone (UVO) (AC-6; AhTech LTS) treated (30 min) Si wafer. The brush-coated Si wafer was thermally annealed at 180 °C under a vacuum for 24 h. The PS-*b*-PMMA were dissolved in toluene (2 wt %), and spin-coated on brush-grafted Si wafer at 3000 rpm for 30 s to fabricate ~61 nm (~2 $L_0$) thick films. Poly(4-tri-methylsilylstyrene)-*b*-poly(4-methoxystyrene), PTMSS-*b*-PMOST (17.9 kg mol⁻¹) and a hydroxyl-terminated neutral brush (poly(4-*tert*-butylstyrene-*co*-3,4-methylenedioxystyrene), P(tBS-*co*-MDOS)−OH) (5.4 kg mol⁻¹) were synthesized by anionic polymerization as described in the previous work[46]. A polarity-switching top coat (poly((maleic anhydride-*alt*-3,5-di-*tert*-butylstyrene)-*co*-(maleic anhydride-*alt*-4-*tert*-butylstyrene))) (57.7 kg mol⁻¹) was synthesized by free radical polymerization. After grafting the neutral brush on Si wafer, 1 wt% PTMSS-*b*-PMOST solution in methyl isobutyl ketone (MIBK) and the top coat solution in methanol were sequentially spin-coated to achieve desired thicknesses. The polydimethylsiloxane (PDMS) elastomer pads were prepared by mixing PDMS elastomer base/curing agent from the Syl-gard 184 (Dow Corning) in a 5:1 ratio and pouring in a 0.8 mm thick mold on a Si wafer. A two-step curing process was executed to form the PDMS pad with nano-bumps, first at 60 °C for 12 h and then at 180 °C for 3 h.

### Rapid shear-rolling process

Unidirectional alignment of the BCP nanopatterns was conducted as follows. First, the stage was preheated to different temperatures from 200 °C to 290 °C, waiting about 10 min to reach thermal equilibrium. 0.8 mm-thick PDMS pads were placed on the BCP films or attached to the roller, then the BCP films were placed on the hot stage. Within 1 min, the distance from the BCP film to the PDMS pad was set, and the shear-rolling process was completed under the optimal conditions of a roller speed of 36 mm·s⁻¹ and a substrate speed of 32 mm·s⁻¹. In order to change the shear rate, we executed the shear-rolling by changing the difference between the two speeds. To control the processing time, we also experimented with changing the substrate speed at the same shear rate. The distance from the BCP film to the PDMS pad was controlled to investigate the influence of normal force on the PDMS pad. All shear-rolling processes were carried out in a single step. Details

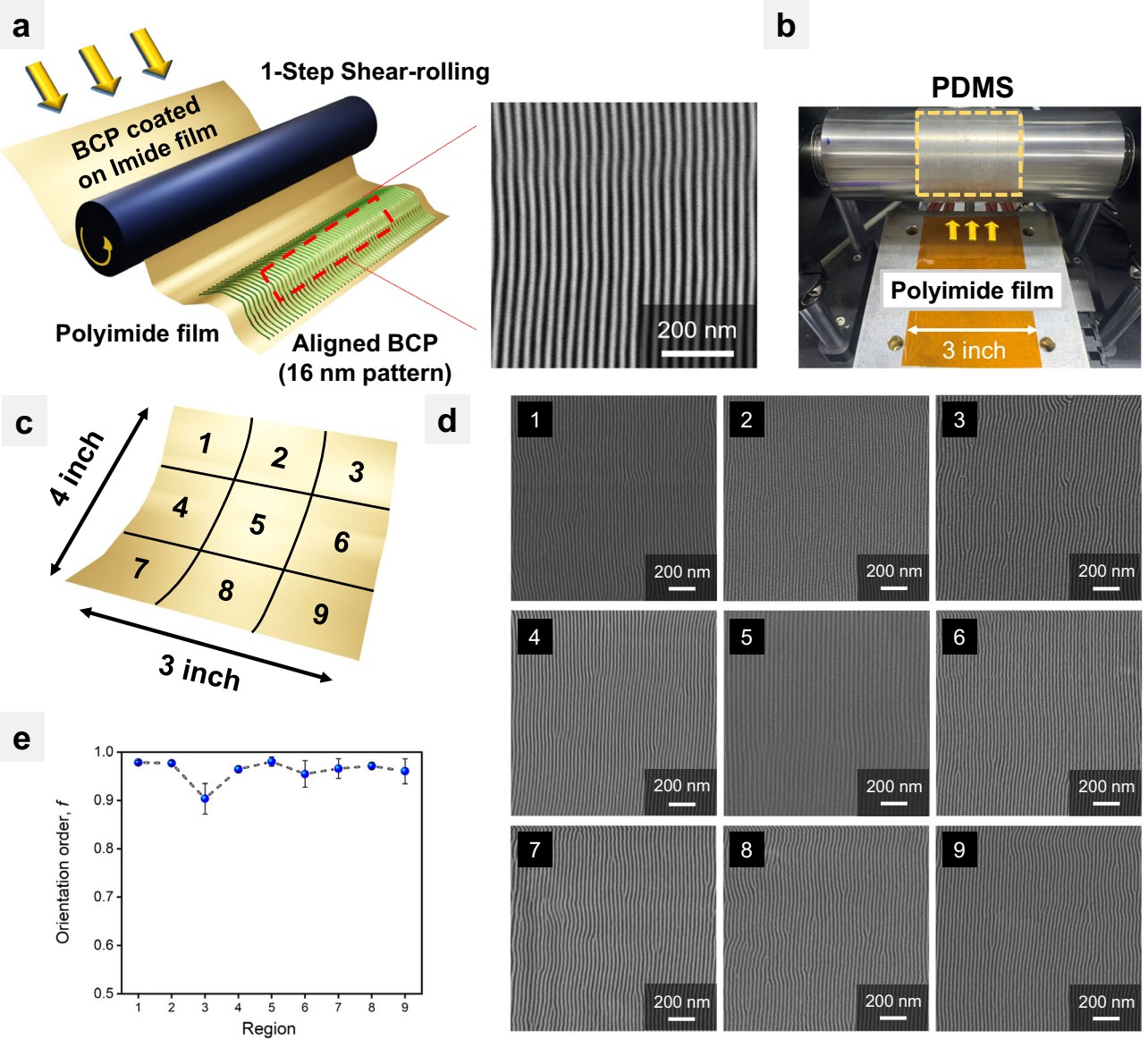

**Fig. 7 | Rapid shear-rolling process with PS-*b*-PMMA film on flexible polyimide substrate for examining roll-to-roll applicability. a** Schematic image of the roll-to-plate 0.1-second rapid shear-rolling on a flexible substrate. **b** A photographic image of 0.1-second rapid shear-rolling on a 3-inch wide long flexible polyimide substrate. **c** Nine regions departed by 32 mm vertically and 26 mm horizontally from the center of the flexible polyimide films. **d** SEM images of rapid shear-rolled PS-*b*-PMMA film on polyimide substrate at designated points. **e** The plot of the orientational order parameters for each region. Source data are provided as a Source Data file.

of the shear-rolling instrument and procedure were also described in our previous study[17].

## Characterization

The thickness of the BCP films was measured using spectral reflectometry (Filmetrics F20, USA). The unidirectional aligned nanopatterns were measured using field-emission scanning electron microscopy (FE-SEM, Zeiss Sigma 300, Zeiss). Before observation with SEM, the BCP films were treated by oxygen plasma (Vita, Femto Science Inc., 20 sccm, 20 mTorr) at 90 W for 10 s. to produce height contrast. For P2VP-*b*-PS-*b*-P2VP film, exposure to iodine vapor for 1 h selectively stained P2VP domains and enhanced contrast in SEM observation. The periodicity and amplitude of the nano-bumpy surface of PDMS were acquired using atomic force microscopy (AFM, XE-7, Park Systems). Image analysis was carried out by using the ImageJ with orientationJ plugin. Grazing Incidence Small-angle X-ray Scattering (GISAXS)

measurements were made on the 9 A beamline of the Pohang Light Source. The energy of the X-ray wavelength was 11.09 keV (wavelength = 1.1179 Å), and the sample-to-detector distance was 2541.6 mm.

## Roll-to-plate shear-rolling with flexible substrate

Commercially available 50 μm thick polyimide film (PI) was used as a flexible substrate. To enhance the grafting density of neutral brushes, the PI surface was treated with UVO for 30 min, following the same treatment used for Si wafers. The resulting PS-*b*-PMMA coated PI film was cut into 3-inch ×4-inch pieces, placed on the hot stage of the shear-rolling machine, and clamped with a vacuum chuck. After very carefully adjusting both sides' height between the roller and the films, the rapid shear-rolling was performed at 280 °C. The BCP nanopatterns on the PI substrate could be observed by SEM through O₂ plasma treatment (90 W, 20 sccm, 20 mTorr, 10 s) and Pt sputtering (10 mA mbar⁻¹, 90 s) sequentially.

## Data availability

The data that support the findings of this study are available from the corresponding author upon request. Source data are provided with this paper.

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

## Acknowledgements

This work was supported by the Korea Institute of Science and Technology (KIST) Institutional and KU-KIST Program (Project No. 2E32501, 2E32503 and 2V09840-23-P025) and the National Research Foundation of Korea (NRF) grant funded by the Korean government (No. NRF-2022R1A2B5B02001597). This research was supported by Basic Science Research Program through the National Research Foundation of Korea(NRF) funded by the Ministry of Education (RS-2023-00273235). In addition, GISAXS experiments at PLS-II 9A beamline of the Pohang Accelerator Laboratory, POSTECH were supported by Dr. Hyungju Ahn and Dr. Wooseop Lee. The authors thank Prof. C. Grant Willson at the University of Texas at Austin for generously supplying us with a silicon-containing, high-χ block copolymer and the associated materials that were synthesized in his lab.

## Author contributions

J.C and J.G.S. conceived the idea and initiated the project. J.C. mainly conducted the experiments and wrote the whole manuscript. J.G.S. supervised the whole project. J.H.K. managed high-χ BCP systems. H.Y.J. and S.C. executed FEM simulation for the stress field. The manuscript was written through the contributions of all authors, including J.O. and J.B. All authors have approved the final version of the manuscript.

## Competing interests

The authors declare no competing interests.
