## [Peer Review File · Nature Communications]

Roll-to-plate 0.1-second shear-rolling process at elevated temperature for highly aligned nanopatternsREVIEWER COMMENTS

Reviewer #1 (Remarks to the Author):

The contribution by Cho et al. describes a variant of existing shear-alignment methods for BCP. While others have shown shear-alignment using PDMS pads as "stress transducers" and the authors have shown in a prior publication the use of a PDMS-coated roller for doing the same; here they show how adding bumpiness to the roller can improve the process. The authors provide empirical studies of the process, showing what regimes optimize quality. There is less in terms of rigorous modeling and thus detailed understanding of the underlying physics. It is impressive that they are able to achieve good order over large areas of material, on flexible substrates amenable to industrial roll-to-roll synthesis. In this sense, they have achieved a record in terms of ease in generating aligned BCP morphologies. This will be quite interesting to the BCP community, but perhaps not to the broader community.

Overall, this is solid work. The technical aspects seem correct, and the text is easy to understand. I believe there are missed opportunities for more rigorous analysis (see below).

A. Stress gradients: The localized contact of the PDMS bumps (which ultimately give rise to the shear forces) presumably leads to a non-trivial shear field in the BCP film. The experimental images look relatively uniform, however, suggesting that this stress field relaxes to be uniform. At first glance, this disparity is surprising. It seems to warrant further investigation. Did the authors consider these points? It seems worthwhile to study the larger-area SEM images, looking for evidence of localized ordering differences that match with where the bumps made contact. Are there any spatial trends in orientation, defectivity, or BCP repeat-spacing that can be observed? Or, can one perform experiments that track exactly where the bumps contacted the film, so that one can measure quantitatively what the order looks like in the vicinity of contact vs. at a distance? An alternate strategy would be to model the stress field one expects for the intermittent contacts. Are the authors claiming that this stress field becomes fully homogenized through the BCP film? (Perhaps due to the high temperature?) If so, evidence should be presented of this.

B. Thermal gradients: From the described methods (page 21, line 400) it sounds that the film temperature was set from the base. I assume the roller was held in place with some kind of arm that was connected to surrounding equipment at room temperature. Thus, I assume the roller is initially at RT while the stage under the BCP is at elevated temperature. This implies that a thermal gradient is established across the film (in the normal direction) and also in the PDMS pad (in the lateral direction) at the contact position. A simple empirical question is whether the results would differ if one instead heated through the roller (keeping the stage at RT) or if one held both roller and stage at high-T (thereby reducing thermal gradients). A deeper question is whether the thermal gradients play a role in the ordering. Previous work has shown that thermal gradients drive differential expansion of PDMS, which gives rise to the shearing in "soft shear" methods. Does that contribute to the observed results? Other work (in the CZA and laser annealing papers) shows that the kinetics of ordering increase as a result of in-plane thermal gradients. Could such gradients be influencing ordering? (E.g. increasing kinetics and helping give rise to the observed order?)

C. Data availability: The authors provide a generic "available upon request" statement. The authors should strongly consider depositing their data (including raw SEM images and x-ray data) to a repository. Making this data broadly available strengthens the disseminated science, and empowers other researchers to more easily build upon their work.

D. Minutia:

D1. The authors call their method "extremely high temperature" but I would call it just "high temperature". Of course these are just vague descriptors. But in the BCP literature, there are some experiments which use temperatures that are in excess of the polymer thermal degradation temperature (~300C), such as RTP (up to 350C reported in their ref 37 as well as related 10.1039/C5TC01473E and 10.1021/acsami.6b02300) or laser methods (500C is reported in ref 27).

I would call annealing temperatures of 180C-220C as "typical", temperatures of 230C-280C as "high" and temperatures of 290-330C as "very high" and temperatures >340C as "extremely high".

D2. page 4, line 66: "the laser-shear method requires an additional heat absorption layer on the substrate." Recent work has shown the laser method can be used with direct substrate absorption (10.1021/acsnano.0c00696).

D3. Figure S3: Why is a line graph being used instead of a bar graph? The x-axis does not represent a continuous variable. Instead of labeling the graph with a, b, c, and d, the authors should simply put the category labels on the graph.

D4. Figure S5: To confirm that the structure in (a) is indeed featureless, the authors could generate an FFT and confirm that there is no peak at the q-spacing of the BCP repeat.

Reviewer #2 (Remarks to the Author):

The paper describes the rapid ordering of sub 10 nm ultrathin block copolymer films by roll-to-roll (R2R) ordering using an ultrahigh processing temperature of 280 C using a moving substrate and roller bumpy PDMS pad with a differential speed for inducing soft shear. The authors determine the optimal conditions for high alignment of the vertical ordering of different block copolymers and expand the work to demonstrate roll-to-roll on a UVO oxidized PI sheet coated with neutral r-PS-PDMS. A high degree of alignment and order is determined for a set of an optimal window of process conditions (contact depth etc.). The contact time is claimed to be on the order of 0.1s for ordering as a rapid ordering process.

While the results are overall interesting and the authors have done a good job of compiling the set of information demonstrating the efficacy of the method in obtaining the aforementioned results, I am not very motivated to recommend the paper for publication in Nature Communication, rather I suggest a more focussed journal. I do not see the broad innovation or exciting scientific accomplishment here, unfortunately.

The reasons for this include:

- 1) Technologically and industrially, processing at ultrahigh 280 C is not so great as it is energy-intensive, and global warming is aiming to process at lower greener temperatures and technologies rather than run chip foundries to higher temperatures.
- 2) Interfacial widths in BCPs (not measured in the paper!) substantially increase with temperature, as also capillary effects due to liquid-like nature of BCP well above T_g , resulting in also Line Edge Roughness (LER), also not measured here! For any real world applications, these go the wrong way with higher temperatures, i.e, broad rather than sharp interfaces, and high LER rather than reduced LER.
- 3) I did not see any major advancement, either in DSA technology or R2R assembly in what is described. Rather, it combines existing and known sets of optimized methods and technology already and optimizes them to bring about the rapid top-down and bottom-up BCP ordering technology

combination. To clarify, let's analyze all the components of what is presented, that were used to obtain the net results: a. 280C High-Temperature + b. Use of Flexible R2R substrates + c. Chemically Patterned Substrate + d. BCP under Differential Roller Soft-Shear

a. Even in static annealing w/o shear, about 1.5 (15 years!) decade ago, it was shown by group of Nealey et al. that 0.3 s annealing @ 280C gave perfect ordering on the same chemically patterned substrate! So the effect of high T, in fact specifically 280 C on BCP annealing to produce ordering in seconds is not new by a long shot. The authors appropriately reference the paper, but give it minimal passing acknowledgment, unfortunately! That paper captures how the thermodynamic and diffusion Activation Energy barrier phenomena should be considered quantitatively for high-temperature ordering. There is no new fundamental science of interest of comparable quality in the present paper along those lines! See, Rapid Directed Assembly of Block Copolymer Films at Elevated Temperatures published in 2008: Adam M. Welander, et al., Juan J. de Pablo, and Paul F. Nealey, *Macromolecules* 2008, 41, 2759-2761. Furthermore, from that paper: "Extrapolation of eq 2 from the data in Figure 2 suggests that very short annealing times may be possible. For example, we would expect annealing times of 9.7 (2.5 and 0.9 (0.3 s at annealing temperatures of 250 and 280 °C), respectively. Indeed, samples annealed at 280 °C for 1 min, the minimum annealing time used in this study, displayed perfect assembly for LS values ranging from 42.5 to 47.5 nm, as shown in Figure 3." Also, insofar as ordering time is concerned, there is comparable time scale ordering of BCP data, e.g. as referenced, Millisecond Ordering of Block Copolymer Films via Photothermal Gradients, *ACS Nano* 2015, 9, 4, 3896–3906

b. There are several published papers on the use of flexible PI and other substrates for R2R BCP ordering, in fact, vertical BCP even on PDMS substrate...

c. Use of chemically patterned random PS-PMMA was pioneered by Thomas P. Russell over 2 decades ago and has been used since then. Along this line notably, there are a number of "template-free" methods, i.e. no chemical pattern, that is preferential/superior, BUT they do not require an additional processing random copolymer step (any additional step in chip processing is highly unfavorable!), e.g. See, Ultra-Fast (3 seconds) Vertical Ordering of Lamellar Block Copolymer Films on Unmodified Substrates, *Macromolecules* 54 (3), 1564-1573;

d. See most recently, for sub-10nm vertical BCP order, (not referenced) modified CZA-SS- Soft-Shear-Aligned Vertically Oriented Lamellar Block Copolymers for Template-Free Sub-10 nm Patterning and Hybrid Nanostructures, *ACS Appl. Mater. Interfaces* 2022, 14, 10, 12824.

4) The only interesting new result I see is the use of bumpy PDMS for soft-shear but here again, the description of how it works and its mechanism is simply "wordy explanations" with no real measurements of contact shear, forces involved etc.

Overall, I think the results are good and interesting, but they do not rise to the level of *Nature Communications*. A more specific journal is recommended.

Reviewer #3 (Remarks to the Author):

The authors report rapid shear rolling at high temperatures (~ 280 °C) to align vertically oriented BCP lamellae in thin films over large areas. The authors show that shear rolling with a bumpy PDMS pad provides macroscopic film stability during shear, resulting in uniformly aligned BCP nanopatterns over the entire area of the film. The authors also optimize the experimental conditions such as normal force, shear rate, and temperature required for highly ordered BCP nanopatterns. Finally, it is demonstrated that the proposed method is also effective for aligning high-chi BCPs and BCPs on different substrates. I think that the manuscript can be published in Nature Communications, but I have a few points which I hope the authors will consider in revision:

(1) The authors claim that the use of bumpy PDMS is essential to prevent macroscopic instability of BCP thin films caused by thermal expansion and contraction of the PDMS pad. Is there any possibility that a preheating at high temperature for 1 minute before the shear application creates a bumpy surface on the flat PDMS pad? I strongly recommend showing an AFM image of a flat PDMS pad after preheating and shearing at high temperature. If this process does not create a bumpy surface, it will support the claim that the stability of the film comes from the premade structure of the PDMS surface.

(2) The authors should clarify the definition of the Hermans' orientation parameter. In addition, theoretical values should be provided for randomly oriented and transversely aligned patterns. Only the value for a perfectly aligned pattern is given in the main text.

(3) I am a bit unclear on the orientation values in Figure 5a and 5e. Although these structures are described as fingerprint-like structures and random orientations in page 13, the orientation parameters are still higher than the theoretical value of a random orientation pattern. How many images did the authors analyze to calculate the orientation order? If the overall orientation is random, the values obtained from sufficient number of images should converge to zero.

(4) Color bars are missing in the orientation and GISAXS 2D maps in Figure 5.

Reviewer #1

Overall comment: The contribution by Cho et al. describes a variant of existing shear-alignment methods for BCP. While others have shown shear-alignment using PDMS pads as "stress transducers" and the authors have shown in a prior publication the use of a PDMS-coated roller for doing the same; here they show how adding bumpiness to the roller can improve the process. The authors provide empirical studies of the process, showing what regimes optimize quality. There is less in terms of rigorous modeling and thus detailed understanding of the underlying physics. It is impressive that they are able to achieve good order over large areas of material, on flexible substrates amenable to industrial roll-to-roll synthesis. In this sense, they have achieved a record in terms of ease in generating aligned BCP morphologies. This will be quite interesting to the BCP community, but perhaps not to the broader community.

Overall, this is solid work. The technical aspects seem correct, and the text is easy to understand. I believe there are missed opportunities for more rigorous analysis (see below).

Reply: We appreciate the reviewer's genuine response. We should continue our nanopatterning research to generate interest within the BCP community and make it universally useful and applicable. As the reviewer pointed out, while we have achieved high-quality results in the BCP nanopatterning field, we may have lacked rigorous analysis. We sincerely appreciate the advice provided during this revision opportunity to help us publish a better-quality paper. Below, we will respond to the comments and questions.

Comment (1-1): A. Stress gradients: The localized contact of the PDMS bumps (which ultimately give rise to the shear forces) presumably leads to a non-trivial shear field in the BCP film. The experimental images look relatively uniform, however, suggesting that this stress field relaxes to be uniform. At first glance, this disparity is surprising. It seems to warrant further investigation. Did the authors consider these points? It seems worthwhile to study the larger-area SEM images, looking for evidence of localized ordering differences that match with where the bumps made contact. Are there any spatial trends in orientation, defectivity, or BCP repeat-spacing that can be observed? Or, can one perform experiments that track exactly where

the bumps contacted the film, so that one can measure quantitatively what the order looks like in the vicinity of contact vs. at a distance? An alternate strategy would be to model the stress field one expects for the intermittent contacts. Are the authors claiming that this stress field becomes fully homogenized through the BCP film? (Perhaps due to the high temperature?) If so, evidence should be presented of this.

Reply: We really appreciate the reviewer's very incisive and helpful comments. In contacting the BCP film and bumpy PDMS pad, non-uniform stress may occur when the nano-bumps of the PDMS are vertically pressed on the BCP film. However, compared to the height of the bump, the scale of the height pressed (deformed) upon contact of the PDMS is much larger than $20\ \mu\text{m}$, so the difference in pressure applied to the film is very small, making it difficult to affect the non-uniform orientation of the BCP film. To support this hypothesis, as recommended in the comments of the reviewers, we executed firstly microscopic SEM and AFM image analysis of a relatively large area, then macroscopic low-mag image analysis, topographical AFM analysis of the BCP film after shear-rolling, and finally stress field FEM simulations.

1. Microscopic SEM and AFM analysis

According to our AFM result in Figure R1, the bumps at the surface of the PDMS pad are observed to have an average diameter of $2\ \mu\text{m}$ and an average spacing of approximately $3.5\ \mu\text{m}$ between them. Figure R2 compares the SEM image (10000x mag, $10\ \mu\text{m} \times 8.5\ \mu\text{m}$) of the shear-rolling treated BCP film and the same-magnification AFM image of the bumpy PDMS pad. The SEM image exhibits a highly aligned line-space pattern with a few dislocation and point defects. However, any micron-sized BCP pattern defects with similar size or period of the bumps on the PDMS were not found. Based on this microscopic result, it was found that the bump did not act as a protrusion during the shear alignment of the nanosized BCP microdomain.

Figure R1. 3D AFM structure and height profile of the bumpy surface of the PDMS pad.

Figure R2. Comparison of SEM image of the shear-rolled BCP film and AFM image of the PDMS pad at the same dimensions (10 μm x 8.5 μm).

2. Macroscopic area analysis (SEM 1000x mag, 114 μm x 85 μm)

In the 1000x magnification SEM image in Figure R3, the bumps of the PDMS pad can be easily observed by treating O_2 plasma to the PDMS surface. However, no morphology could be observed in the SEM image of the BCP film treated with shear-rolling at the same magnification.

Figure R3. Comparison of SEM images of the bumpy PDMS pad and shear-rolling treated BCP film at the same dimensions (114 μm x 85 μm).

3. AFM result of the BCP film after shear-rolling

To the reviewer's concern, if the bump structure creates local stress fields on the BCP film surface, traces of the bump may remain like stamping when shear stress is applied to the BCP film. Therefore, topographical AFM analysis of the BCP film surface was investigated to check the occurrence of craters by bumps with a height of about 20 nm. However, as shown in Figure R4, the topographical AFM images of shear-rolling treated BCP film show very flat morphology with 0.4 nm of root-mean-square roughness (R_q). This result supports the validity of our claim that the bumps first apply a relatively averaged stress to the BCP film due to shear stress before imparting localized stress, as we expected.

Figure R4. AFM topographical and phase images of shear-rolling treated BCP film.

Correction: Sentences related to the formation of the shear field analysis through SEM, AFM images, and simulation results has been included on page 7, line 145. Additionally, Figure R2 and Figure R4 have been newly included as Figure S1 and Figure S2 in the supporting information.

Page 7, line 142:

“The surface of the BCP film did not show any stamping traces caused by the bumpy PDMS structure (medium magnification of SEM image in Figure S1 and AFM topographical image in Figure S2). In addition, by modeling the distribution of the stress field during intermittent contact in Figure S4 in supporting information, simulation results confirm that a uniform shear stress can be applied when such contact occurs.”

Figure S1. Comparison of SEM and AFM data with the same dimensions ($10\ \mu\text{m} \times 8.5\ \mu\text{m}$). (a) SEM image of shear-rolled BCP film. (b) AFM image of bumpy PDMS. A non-trivial shear field does not form at a bump location.

Figure S2. AFM results of shear-rolled BCP film surface. ($5\ \mu\text{m} \times 5\ \mu\text{m}$). The film's surface roughness ($R_q = 0.428\ \text{nm}$) of the BCP film after shear-rolling indicates that when bumps apply

shear stress to the BCP film, a homogeneous interface is first formed and then the stress is transferred. (height of bump: 25 nm)

4. FEM Simulation

We have additionally confirmed simulations of the stress field distribution that may occur during the contact between PDMS and the BCP film. In this simulation, we assumed a linear elastic model simply to understand the impact of bumps on the stress field of the BCP film. At high temperatures, the modulus of the BCP film is notably low, allowing it to exist in a mobile state characteristic of the rubbery regime. Therefore, we conducted simulations with the BCP film's modulus minimized to its lowest possible value. The simulations to investigate the stress distribution when normal pressure is applied to the surface of the BCP film with a $10\ \mu\text{m} \times 10\ \mu\text{m}$ area of PDMS with four bump structures. Based on the trends observed in this normal stress distribution, we predicted the distribution of shear stress. These simulations were performed for four different conditions, increasing the pressing depth from 5 nm to 155 nm, each increment being 50 nm. When pressed 5 nm, which is less than the length of the bump (~ 20 nm), a localized stress field can be observed due to the bump structure (Figure R5a). Even at a pressing depth of 55 nm, the traces of the bumps are clearly visible (Figure R5b). However, as greater pressure is applied to a deeper extent, the localized stress field dissipates into the geometric deformation of the PDMS surface which can lead to a fully homogenized stress field.

Figure R5. FEM simulation results of stress field on the BCP film with different contact displacements. (a) Pressing depth = 5 nm, (b) 55 nm, (c) 105 nm, (d) 155 nm. The unit of z displacement is nm, and Von mises stress is Pa.

In our shear-rolling process, applying pressing depth with 20 μm in the vertical direction leads us to anticipate an even more uniform stress field formation.

To validate our simulation results, we conducted a simple pressing experiment. To minimize the preferential wetting-induced terrace structures and to focus solely on the physical effects caused by the nano-bumps, we executed the PDMS pressing on thicker BCP film (~ 300 nm) than our experimental conditions (~ 62 nm). Figure R6 shows the AFM images of pristine PS-*b*-PMMA films before and after applying pressure with 20 μm of deformation in the vertical direction. In Figure R6a, the pristine PS-*b*-PMMA film exhibits a relatively flat surface from the spin coating. However, after applying pressing depth with 20 μm at 280 $^{\circ}\text{C}$ for 1 s, the film did not leave traces of bumpy structure and rather formed a flatter surface. These results, consistent with simulations, indicate that the BCP films undergo flattening deformation rather than stamping of the bumpy structure when mild pressure is applied, even at high temperatures. Based on these results, it can be concluded that the PDMS pad with nano-bumps can achieve conformal contact and uniform shear transfer to the BCP film if sufficient pressure is applied during the shear-rolling, and can be separated immediately after shear-rolling.

Figure R6. AFM results of PS-*b*-PMMA film surface (a) before and (b) after pressing with 20 μm deformation at 280 $^{\circ}\text{C}$ for 1 s.

The details of the simulation conditions are below.

The stress field on the BCP film was calculated using the FEM simulation tool (structural module of COMSOL Multiphysics). The PDMS was modeled as the size of $10\ \mu\text{m} \times 10\ \mu\text{m} \times 1\ \mu\text{m}$ rectangular shape. The bump structures were modeled as elliptical shapes with a peak height of 25 nm and a diameter of $2\ \mu\text{m}$. Four Bump structures were allocated on the PDMS surface with an average distance of $3.5\ \mu\text{m}$. The BCP film was modeled as the size of $20\ \mu\text{m} \times 20\ \mu\text{m} \times 60\ \text{nm}$ rectangular shape and was placed with the distance of 500 nm from the PDMS surface. Young's modulus of material was set as 750 kPa for the PDMS and 1 MPa for the BCP film and their Poisson's ratio was set as 0.49 and 0.35. Bumpy surface of PDMS and the top surface of the BCP film were set as contact pair. The bottom surface of the BCP film was designated as the fixed boundary condition. The top surface of the PDMS structure was set as prescribed displacement condition to make a contact with the BCP film. Other boundaries are set as free to allow structural deformation. The contact analysis was performed with the desired contact situation.

The above simulation results are incorporated into our supporting data.

Correction: In supplementary information, Figure R5 and R6 have been newly added as **Figure S3** and **Figure S4**, respectively and explanation of the stress field distribution part.

FEM simulation of stress field distribution

The stress field on the BCP film was calculated using the FEM simulation tool (structural module of COMSOL Multiphysics). The PDMS was modeled as the size of $10\ \mu\text{m} \times 10\ \mu\text{m} \times 1\ \mu\text{m}$ rectangular shape. The bump structures were modeled as elliptical shapes with a peak height of 25 nm and a diameter of $2\ \mu\text{m}$. Four Bump structures were allocated on the PDMS surface with an average distance of $3.5\ \mu\text{m}$. The BCP film was modeled as the size of $20\ \mu\text{m} \times 20\ \mu\text{m} \times 60\ \text{nm}$ rectangular shape and was placed with the distance of 500 nm from the PDMS surface. Young's modulus of material was set as 750 kPa for the PDMS and 1 MPa for the BCP

film and their Poisson's ratio was set as 0.49 and 0.35. Bumpy surface of PDMS and the top surface of the BCP film were set as contact pair. The bottom surface of the BCP film was designated as the fixed boundary condition. The top surface of the PDMS structure was set as prescribed displacement condition to make a contact with the BCP film. Other boundaries are set as free to allow structural deformation. The contact analysis was performed with the desired contact situation.

In this simulation, we assumed a linear elastic model simply to understand the impact of bumps on the stress field of the BCP film. At high temperatures, the modulus of the BCP film is notably low, allowing it to exist in a mobile state characteristic of the rubbery regime. Therefore, we conducted simulations with the BCP film's modulus minimized to its lowest possible value. The simulations to investigate the stress distribution when normal pressure is applied to the surface of the BCP film with a $10\ \mu\text{m} \times 10\ \mu\text{m}$ area of PDMS with four bump structures. Based on the trends observed in this normal stress distribution, we predicted the distribution of shear stress. These simulations performed for four different conditions, increasing the pressing depth from 5 nm to 155 nm, each increment being 50 nm. When pressed 5 nm, which is less than the length of the bump ($\sim 20\ \text{nm}$), localized stress field can be observed due to the bump structure (Figure S4 (a)). Even at a pressing depth at 55 nm, the traces of the bumps are clearly visible (Figure S4 (b)). However, as greater pressure is applied to a deeper extent, the localized stress field dissipated into the geometric deformation of PDMS surface which can lead to fully homogenized stress field. In our shear-rolling process, applying pressing depth with $20\ \mu\text{m}$ in the vertical direction leads us to anticipate an even more uniform stress field formation.

To validate our simulation results, we conducted a simple pressing experiment. To minimize the preferential wetting-induced terrace structures and to focus solely on the physical effects

caused by the nano-bumps, we executed the PDMS pressing on thicker BCP film (~300 nm) than our experimental conditions (~62 nm). Figure R6 shows the AFM images of pristine PS-b-PMMA films before and after applying pressure with 20 μm of deformation in the vertical direction. In Figure R6a, the pristine PS-b-PMMA film exhibits a relatively flat surface from the spin coating. However, after applying pressing depth with 20 μm at 280 $^{\circ}\text{C}$ for 1 s, the film did not leave traces of bumpy structure and rather formed a flatter surface. These results, consistent with simulations, indicate that the BCP films undergo flattening deformation rather than stamping of the bumpy structure when mild pressure is applied, even at high temperatures. Based on these results, it can be concluded that the PDMS pad with nano-bumps can achieve conformal contact and uniform shear transfer to the BCP film if sufficient pressure is applied during the shear-rolling, and can be separated immediately after shear-rolling.

Figure S3. FEM simulation results of stress field on the BCP film with different contact displacements. (a) Pressing depth = 5 nm, (b) 55 nm, (c) 105 nm, (d) 155 nm. The unit of z displacement is nm, and Von mises stress is Pa.

Figure S4. AFM results of PS-b-PMMA film surface (a) before and (b) after pressing with 20 μm deformation at 280 $^{\circ}\text{C}$ for 1 s.

Comment (1-2): B. Thermal gradients: From the described methods (page 21, line 400) it sounds that the film temperature was set from the base. I assume the roller was held in place with some kind of arm that was connected to surrounding equipment at room temperature. Thus, I assume the roller is initially at RT while the stage under the BCP is at elevated temperature. This implies that a thermal gradient is established across the film (in the normal direction) and also in the PDMS pad (in the lateral direction) at the contact position. A simple empirical question is whether the results would differ if one instead heated through the roller (keeping the stage at RT) or if one held both roller and stage at high-T (thereby reducing thermal gradients). A deeper question is whether the thermal gradients play a role in the ordering. Previous work has shown that thermal gradients drive differential expansion of PDMS, which gives rise to the shearing in "soft shear" methods. Does that contribute to the observed results? Other work (in the CZA and laser annealing papers) shows that the kinetics of ordering increase as a result of in-plane thermal gradients. Could such gradients be influencing ordering? (E.g.

increasing kinetics and helping give rise to the observed order?)

Reply: As suggested by the reviewer, we have investigated whether thermal gradients could influence the alignment. To reduce the vertical thermal gradient, the PDMS pad was placed on the surface of the BCP film, pre-annealed for 5 minutes on hot-plate to achieve thermal equilibrium, and then shear-rolled. In particular, since the heat transfer coefficient of PDMS is much lower (and thicker!) than that of PS and PMMA, we think that the temperature gradient in the vertical direction within the 50 nm thick BCP film in this experiment is negligible. This experiment was conducted in the P2VP-*b*-PS-*b*-P2VP system using a filter plasma capable of top neutralization even for the PDMS pad because it was difficult to form the top neutrality of PS-*b*-PMMA in long-time contact with PDMS. From the pre-annealed BCP film in contact with PDMS for a long time in Figure R7, it is clearly confirmed that effective alignment is achieved due to shear stress even in the absence of a thermal gradient.

Figure R7. SEM images of after shear-rolling at 230 °C with the conditions of (a) the PDMS pad on roller (far from hot plate) (b) the PDMS pad on BCP film (thermally equilibrium condition)

Moreover, we can demonstrate the impact of shear rate on alignment by controlling the roller velocity using PS-*b*-PMMA. By adjusting shear stress, significant variations in alignment can be observed as Figure R8. As a result, we can claim that in our alignment experiments, shear stress plays a more dominant role as a significant factor compared to thermal gradients.

Figure R8. SEM images after shear-rolling at different shear rates. (a) $v_x = 36$ mm/s, $v_r = 36$ mm/s (b) $v_x = 36$ mm/s, $v_r = 40$ mm/s. Processing temperatures are 280 °C.

Comment (1-3): C. Data availability: The authors provide a generic "available upon request" statement. The authors should strongly consider depositing their data (including raw SEM images and x-ray data) to a repository. Making this data broadly available strengthens the disseminated science, and empowers other researchers to more easily build upon their work.

Reply: We completely agree with the reviewer's feedback. In order to uphold the transparency of our research findings and contribute to the advancement of the scientific community, we will make our essential SEM images and GISAXS raw data publicly available.

Comment (1-4): D. Minutia: D1. The authors call their method "extremely high temperature" but I would call it just "high temperature". Of course these are just vague descriptors. But in the BCP literature, there are some experiments which use temperatures that are in excess of the polymer thermal degradation temperature (~300C), such as RTP (up to 350C reported in their ref 37 as well as related 10.1039/C5TC01473E and 10.1021/acsami.6b02300) or laser methods (500C is reported in ref 27). I would call annealing temperatures of 180C-220C as "typical", temperatures of 230C-280C as "high" and temperatures of 290-330C as "very high" and temperatures >340C as "extremely high".

Reply: Thank you for commenting on an expression commonly used in the RTP research community for temperature. As reviewer's suggestion, we have revised the expression "**extremely high temperature**" to "**high temperature**". page 2, line 23; page 5, line 98; page 6, line 115 right figure; page 6, line 117; page 14, line 273 (extremely high temperature to **very high temperature** (290 °C).

D2. Page 4, line 66: "the laser-shear method requires an additional heat absorption layer on the substrate." Recent work has shown the laser method can be used with direct substrate absorption (10.1021/acsnano.0c00696).

Reply: As reviewer's comments, we have included additional content related to the laser-shear method and added the relevant reference.

page 4, line 66; "On the other hand, the laser-shear method requires an additional heat absorption layer **or base temperature controls²⁹ on the substrate to provide appropriate mobility to BCP chains.**"

Additional reference at page 4:

(29) Leniart, A. A., Pula, P., Sitkiewicz, A. & Majewski, P. W. Macroscopic Alignment of Block Copolymers on Silicon Substrates by Laser Annealing. *ACS Nano* **14**, 4805–4815 (2020).

D3. Figure S3: Why is a line graph being used instead of a bar graph? The x-axis does not represent a continuous variable. Instead of labeling the graph with a, b, c, and d, the authors should simply put the category labels on the graph.

Reply: Thanks for your advice for the graph type. As reviewer's comments, we have revised the graph type from a line to a bar graph in Figure S3, and also categorized the labels on the graph.

Correction: Figure S3 → Figure S9

Figure S9. Orientation order graphs according to the surface defect type. (a) Flat, (B) Partially undulated, (c) Undulated, (d) Undulated and severe ridged surface, respectively.

D4. Figure S5: To confirm that the structure in (a) is indeed featureless, the authors could generate an FFT and confirm that there is no peak at the q-spacing of the BCP repeat.

Reply: We appreciate your advice about the FFT analysis. As reviewer's comments, we have added the FFT image besides of the SEM images in Figure S5. The parallel oriented BCP patterns does not show any peaks in the FFT image.

Correction: Figure S5 → Figure S11, and add the FFT results.

Figure S11. SEM images of thermal annealing w/ and w/o PDMS on the top surface of BCP film. (a) The PS-b-PMMA film in contact with the PDMS pad for only first 2 seconds of the thermal annealing at 280 °C formed a parallel orientation from the top surface. (b) The perpendicular oriented structure of PS-b-PMMA after thermal annealing at 280 °C. The thermal annealing time is equal to 1 min for both (a) and (b). Images (c) and (d) respectively show the FFT results of (a) and (b).

Reviewer #2

Overall comment: The paper describes the rapid ordering of sub 10 nm ultrathin block copolymer films by roll-to-roll (R2R) ordering using an ultrahigh processing temperature of 280 C using a moving substrate and roller bumpy PDMS pad with a differential speed for inducing soft shear. The authors determine the optimal conditions for high alignment of the vertical ordering of different block copolymers and expand the work to demonstrate roll-to-roll on a UVO oxidized PI sheet coated with neutral r-PS-PDMS. A high degree of alignment and order is determined for a set of an optimal window of process conditions (contact depth etc.). The contact time is claimed to be on the order of 0.1s for ordering as a rapid ordering process. While the results are overall interesting and the authors have done a good job of compiling the set of information demonstrating the efficacy of the method in obtaining the aforementioned results, I am not very motivated to recommend the paper for publication in Nature Communication, rather I suggest a more focused journal. I do not see the broad innovation or exciting scientific accomplishment here, unfortunately. The reasons for this include:

Reply: First of all, thank you for your good feedback, and we regret that our paper did not make the strong impression to reviewer #2. However, we believe that the results of this study can generate high-aspect-ratio sub-10 nm patterns with unidirectional alignment across a large area within a short time even on flexible substrates, thereby increasing the applicability of Directed Self-Assembly (DSA)-based nanolithography to various fields, not limited to the semiconductor. Even in the large-scale roll-to-roll film process used in the industry, it is unique and has universal applicability in that highly aligned BCP nanopatterns can be produced in a single shear by varying the speed of the rollers. In particular, in order to manufacture a waveguide for an EUV light source used in EUV lithography, a unidirectionally aligned line-space patterns with tens of nanometer width is required over a large area of 12 inches or more, but it is very difficult to manufacture it. As a process that can produce this, 0.1 sec shear-rolling (0.1sSR) can be considered. In addition, the unidirectional orientation of the block copolymer in a large area can impart the mechanical anisotropy of the block copolymer-based thermoplastic elastomer without any optical interference such as diffraction of visible light, thus giving various functionalities to the elastomer and stretchable electronics.

However, this large-scale orientation process is difficult to implement except for applying mechanical external force, and even in the process of applying mechanical shear, it was not

possible to successfully achieve it because of the very sensitive condition that the nanostructure should be nicely aligned during shearing while avoiding macroscopic unstable movements like undulation or delamination at the interface. In this paper, shear-rolling was applied sequentially for 0.1 seconds to successfully orient nanoscale structures while preventing large-scale delamination through several strategies, thereby achieving high-quality unidirectional orientation for the first time at an ultra-high temperature of 280 °C despite using mechanical external forces. In this regard, we believe that this result has a significant impact on the field of directed self-assembly nanopatterning and broaden its potential for practical industrial applications as it enables rapid and large-scale nanostructure orientation in an industrial-scale roll-to-roll process. We would like to address these concerns and seek your concurrence through the detailed responses provided below.

Comment (2-1): Technologically and industrially, processing at ultrahigh 280 C is not so great as it is energy-intensive, and global warming is aiming to process at lower greener temperatures and technologies rather than run chip foundries to higher temperatures.

Reply: I agree with your opinion in general terms. However, in the semiconductor process or thin film process, the rapid thermal annealing (RTA) process is widely used because all equipment does not go to high temperatures, only the substrate is quickly heated, and then cools down quickly after the desired process, so relatively energy consumption is not so great. In the case of self-assembly of block copolymers, it takes a lot of time to form an ordered structure unless it is a high-temperature process, and only batch processing is possible. So imec and other institutes for trying to combine EUV lithography and DSA are also conducting high temperature annealing at 280 °C for 5 minutes. Since our shear-rolling can be implemented with only a contact time of 0.1 second at high temperature, a large-area continuous process becomes possible, and the resulting cost reduction will be much more valuable than the cost incurred for high temperature.

Comment (2-2): Interfacial widths in BCPs (not measured in the paper!) substantially increase with temperature, as also capillary effects due to liquid-like nature of BCP well above T_g , resulting in also Line Edge Roughness (LER), also not measured here! For any real world applications, these go the wrong way with higher temperatures, i.e, broad rather than sharp

interfaces, and high LER rather than reduced LER.

Reply: We appreciate the reviewer's comment. As pointed out by the reviewer, an increase in temperature can lead to a decrease in the χ value, which may result in an increase in interfacial width and resulting Line edge roughness (LER). However, when calculating the LER, it does not increase significantly to 3.7 nm at 280 °C compared to 3.5 nm at a low temperature of 240 °C. This is mainly attributed to the highly aligned nanopattern that was formed at high temperature (280 °C) (Figure R9). Additionally, we also successfully demonstrated unidirectional alignment for high- χ BCPs (P2VP-*b*-PS-*b*-P2VP, PTMSS-*b*-PMOST) as a solution to this issue. The LER data of line-space nanopattern was obtained by selecting a region of interest (ROI) then calculated using Lacerm (Line and Contact Edge Roughness Meter) program written in Matlab code. This program was developed by Dr. Cong Que Dinh and is available for use at www.lacerm.com.

Figure R9. Comparison of Line-edge roughness (LER) values at 240 °C and 280 °C using Lacerm program.

In accordance with the comments from the reviewer, we have newly added the above LER

data (Figure R9) in the supporting information section (as referred Figure S12) and added related sentences in the manuscript.

Correction:

Page 14, line 265:

“This alignment results in increased order parameters to over 0.97 and low line edge roughness (LER) of 3.5 nm (Figure S12), although some defects may still be present.”

Page 14, line 270:

“Also, the LER did not increase significantly to 3.7 nm.”

Figure S12. Comparison of Line-edge roughness (LER) values at 240 °C and 280 °C. LER is defined as 3 times its standard deviation σ . The LER data of line-space nanopattern was obtained by selecting a region of interest (ROI) then calculated using Lacerm (Line and Contact Edge Roughness Meter) program written in Matlab code. This program was developed by Dr. Cong Que Dinh and is available for use at www.lacerm.com.

Comment (2-3): I did not see any major advancement, either in DSA technology or R2R assembly in what is described. Rather, it combines existing and known sets of optimized methods and technology already and optimizes them to bring about the rapid top-down and bottom-up BCP ordering technology combination. To clarify, let's analyze all the components of what is presented, that were used to obtain the net results: a. 280C High-Temperature + b. Use of Flexible R2R substrates + c. Chemically Patterned Substrate + d. BCP under Differential Roller Soft-Shear

a. Even in static annealing w/o shear, about 1.5 (15 years!) decade ago, it was shown by group of Nealey et al. that 0.3 s annealing @ 280C gave perfect ordering on the same chemically patterned substrate! So the effect of high T, in fact specifically 280 C on BCP annealing to produce ordering in seconds is not new by a long shot. The authors appropriately reference the paper, but give it minimal passing acknowledgment, unfortunately! That paper captures how the thermodynamic and diffusion Activation Energy barrier phenomena should be considered quantitatively for high-temperature ordering. There is no new fundamental science of interest of comparable quality in the present paper along those lines! See, Rapid Directed Assembly of Block Copolymer Films at Elevated Temperatures published in 2008: Adam M. Welfander, et al., Juan J. de Pablo, and Paul F. Nealey, *Macromolecules* 2008, 41, 2759-2761. Furthermore, from that paper: "Extrapolation of eq 2 from the data in Figure 2 suggests that very short annealing times may be possible. For example, we would expect annealing times of 9.7 (2.5 and 0.9 (0.3 s at annealing temperatures of 250 and 280 °C), respectively. Indeed, samples annealed at 280 °C for 1 min, the minimum annealing time used in this study, displayed perfect assembly for LS values ranging from 42.5 to 47.5 nm, as shown in Figure 3." Also, insofar as ordering time is concerned, there is comparable time scale ordering of BCP data, e.g. as referenced, Millisecond Ordering of Block Copolymer Films via Photothermal Gradients, *ACS Nano* 2015, 9, 4, 3896–3906.

b. There are several published papers on the use of flexible PI and other substrates for R2R BCP ordering, in fact, vertical BCP even on PDMS substrate...

c. Use of chemically patterned random PS-PMMA was pioneered by Thomas P. Russell over 2 decades ago and has been used since then. Along this line notably, there are a number of "template-free" methods, i.e. no chemical pattern, that is preferential/superior, BUT they do not require an additional processing random copolymer step (any additional step in chip processing is highly unfavorable!), e.g. See, Ultra-Fast (3 seconds) Vertical Ordering of Lamellar Block Copolymer Films on Unmodified Substrates, *Macromolecules* 54 (3), 1564-

1573;

d. See most recently, for sub-10nm vertical BCP order, (not referenced) modified CZA-SS-Soft-Shear-Aligned Vertically Oriented Lamellar Block Copolymers for Template-Free Sub-10 nm Patterning and Hybrid Nanostructures, ACS Appl. Mater. Interfaces 2022, 14, 10, 12824.

Reply: Thanks for the reviewer's comment that it's a combination of the existing technologies. However, I have a hard time agreeing with that opinion and think we've made a major achievement that researchers have not been able to achieve so far.

(for a) Clearly, **self-assembly at high temperatures of 280 °C**, forming energetically stable nanostructures in minutes, **should be developed for rapid and large-scale patterning process**. In chemoepitaxy, as known as LiNe flow in industrial DSA lithography field, which is oriented along the chemical pattern below, the process at high temperature has been established as the reviewer already mentioned, and imec is currently performing annealing process of BCP films at 280 °C for 5 min as a routine in the conventional 12-inch wafer patterning process.

However, in the approach of applying a **mechanical external force** such as shear force for orientation of the BCP nanostructure, when the mobility of the polymer chain increases at high temperatures, the nanostructure can be well-oriented, but **dewetting or delamination of the BCP film due to mass migration** at the shear-applied interface can occur. So, until now, mechanical shearing-assisted nanodomain orientation of BCP thin films at a high temperature over 200 °C has not been performed, and the degree of orientation has been only improved through static shear for a long time around 30 min or repetitive shearing.

In this paper, by **sequentially applying shear-rolling for 0.1 seconds** to only orient the nanoscale structure successfully but prevent large-scale delamination, **we firstly achieved high-quality unidirectional orientation at ultra-high temperature of 280 °C even with mechanical external force**. In particular, as strategies to minimize instability at the interface, we introduced a **nano-bump structure on the pad** so that the substrate and the PDMS pad do not contact except when shear is applied, and **minimized the contact time and contact area**.

(for b) If this becomes possible, simply by providing a speed difference between the two rolls in the roll-to-roll process, shearing can be successfully applied to the BCP on the flexible substrate. And then, in the roll-to-roll process, unidirectionally aligned nanopatterns on large area flexible substrate can be realized. **We demonstrated roll-to-plate directionally aligning process like the roll-to-roll on a 3-inch-wide continuous polyimide film**. However, **no one has achieved unidirectional orientation of nanostructures through a continuous process**

on a flexible substrate. There are only studies on forming nanostructures of BCP films on flexible substrates, not directional alignment. In this regard, this technology will be an innovative basis for new industrial approaches in applications that require unidirectionally aligned nanopatterns but want to be implemented in large-area films rather than the semiconductor on-chip process.

(for c and d) For the neutrality, there have been recent studies on automatic vertical alignment. However, **device-friendly unidirectional aligned pattern processes for DSA such as chemoepitaxy or shear alignment result in the formation of numerous defects unless the perfect neutrality condition for perpendicular orientation.** In the case of the shear that makes physical contact with PDMS and moves the polymer chain by applying mechanical force, due to the very low surface energy of PDMS and the movement of many chains, most of the BCP films cannot maintain a metastable perpendicular structure and transform to the most energetically stable parallel orientation. So, for the formation of perpendicular orientation and directional alignment at the same time in the **shear alignment processes, it is necessary to introduce neutral layers on top and at the bottom of the BCP films, which is also well shown in the soft-shear paper presented by the reviewer in d.** The latest soft shear paper on simultaneous vertical and unidirectional alignment, commented by the reviewer in d, *ACS Appl. Mater. Interfaces* **2022**, *14*, 12824, is additionally cited in [24] of the revised manuscript.

Correction: Page 3, line 52; Page 4, line 62; reference 24 is newly added to the text below.

Page 3, line 53:

“In order to overcome these challenges, different strategies for DSA have been developed, including graphoepitaxy,⁸⁻¹⁰ chemoepitaxy¹¹⁻¹⁶ and shear alignment¹⁷⁻²⁴ methods.”

Page 4, line 63:

"Cold zone annealing with soft-shear (CZA-SS)^{22,24} and laser-zone annealing²⁶⁻²⁸ are sequential shear alignment methods by localized and directional thermal expansion of the PDMS pad using intense thermal gradients generated by highly localized heating.”

Additional reference at page 3, 4:

(24) Singh, M. et al. Soft-Shear-Aligned Vertically Oriented Lamellar Block Copolymers for Template-Free Sub-10 nm Patterning and Hybrid Nanostructures. *ACS Appl. Mater. Interfaces*

14, 12824–12835 (2022).

4) The only interesting new result I see is the use of bumpy PDMS for soft-shear but here again, the description of how it works and its mechanism is simply “wordy explanations” with no real measurements of contact shear, forces involved etc.

Reply: Thanks for the comment about wordy explanations. We also newly analyzed through experimental investigation and FEM simulation why the shear-rolled BCP film with bumpy PDMS was exhibited microscopically flat morphologies without any macroscopic stamping. Below are the revised paper in the manuscript for this:

First, we wanted to confirm whether there were any additional effects that we had not taken into account on the BCP film when applied to shear-rolling process using PDMS with a bumpy structure, so we compared the AFM image of the bumps, along with their dimensions, to the SEM image to verify if any defects in the nanopattern occurred at the bump's location. Figure R10 compares the SEM image (10000x mag, 10 μm x 8.5 μm) of the shear-rolling treated BCP film and the same-magnification AFM image of the bumpy PDMS pad. The SEM image exhibits a highly aligned line-space pattern with a few dislocation and point defects. However, any micron-sized BCP pattern defects with similar size or period of the bumps on the PDMS were not found. Based on this microscopic result, it was found that the bump did not act as a protrusion during the shear alignment of the nanosized BCP microdomain.

Furthermore, if the bump structure creates local stress fields on the BCP film surface, traces of the bump may remain like stamping when shear stress is applied to the BCP film. Therefore, topographical AFM analysis of the BCP film surface was investigated to check the occurrence of craters by bumps with a height of about 20 nm. However, as shown in Figure R11, the topographical AFM images of shear-rolling treated BCP film show very flat morphology with 0.4 nm of root-mean-square roughness (R_q). This result supports the validity of our claim that the bumps first apply a relatively averaged stress to the BCP film due to shear stress before imparting localized stress, as we expected.

We also conducted simulations and additional experiments to gain a more detailed understanding of how the stress distribution would occur in the BCP film when shear-rolling with PDMS containing the bump structure. In this simulation, we assumed a linear elastic model simply to understand the impact of bumps on the stress field of the BCP film. At high

Figure R10. Comparison of SEM image of the shear-rolled BCP film and AFM image of the PDMS pad at the same dimensions (10 μm x 8.5 μm).

temperatures, the modulus of the BCP film is notably low, allowing it to exist in a mobile state characteristic of the rubbery regime. Therefore, we conducted simulations with the BCP film's modulus minimized to its lowest possible value. The simulations to investigate the stress distribution when normal pressure is applied to the surface of the BCP film with a $10\ \mu\text{m} \times 10\ \mu\text{m}$ area of PDMS with four bump structures. Based on the trends observed in this normal stress distribution, we predicted the distribution of shear stress. These simulations performed for four different conditions, increasing the pressing depth from 5 nm to 155 nm, each increment being 50 nm. When pressed 5 nm, which is less than the length of the bump ($\sim 20\ \text{nm}$), localized stress field can be observed due to the bump structure (Figure R12a). Even at a pressing depth at 55 nm, the traces of the bumps are clearly visible (Figure R12b). However, as greater pressure is applied to a deeper extent, the localized stress field dissipated into the geometric deformation of PDMS surface which can lead to fully homogenized stress field. In our shear-rolling process, applying pressing depth with $20\ \mu\text{m}$ in the vertical direction leads us to anticipate an even more uniform stress field formation.

Figure R11. AFM topographical and phase images of shear-rolling treated BCP film.

Figure R12. FEM simulation results of stress field on the BCP film with different contact displacements. (a) Pressing depth = 5 nm, (b) 55 nm, (c) 105 nm, (d) 155 nm. The unit of z displacement is nm, and Von mises stress is Pa.

To validate our simulation results, we conducted a simple pressing experiment. To minimize the preferential wetting-induced terrace structures and to focus solely on the physical effects caused by the nano-bumps, we executed the PDMS pressing on thicker BCP film (~ 300 nm) than our experimental conditions (~ 62 nm). Figure R13 shows the AFM images of pristine PS-b-PMMA films before and after applying pressure with $20\ \mu\text{m}$ of deformation in the vertical direction. In Figure R12a, the pristine PS-b-PMMA film exhibits a relatively flat surface from the spin coating. However, after applying a $20\ \mu\text{m}$ depth of pressure at $280\ ^\circ\text{C}$ for 1 s, the film did not leave traces of bumpy structure and rather formed a flatter surface. These results, consistent with simulations, indicate that the BCP films undergo flattening deformation rather than stamping of the bumpy structure when mild pressure is applied, even at high temperatures. Based on these results, it can be concluded that the PDMS pad with nano-bumps can achieve conformal contact and uniform shear transfer to the BCP film if sufficient pressure is applied during the shear-rolling, and can be separated immediately after shear-rolling.

Figure R13. AFM results of PS-*b*-PMMA film surface (a) before and (b) after pressing with 20 μm deformation at 280 $^{\circ}\text{C}$ for 1 s.

Thanks to the important comments from the reviewer, we were able to provide a more detailed explanation of the process mechanism of our experiment. These contents will be added to the manuscript and supporting information.

Correction: Sentences related to the formation of the shear field analysis through SEM, AFM images, and simulation results has been included on page 7, line 145. Additionally, Figure R10 and Figure R11 have been newly included as Figure S1 and Figure S2 in the supporting information.

Page 7, line 143:

“The surface of the BCP film did not show any stamping traces caused by the bumpy PDMS structure (medium magnification of SEM image in Figure S1 and AFM topographical image in Figure S2). In addition, by modeling the distribution of the stress field during intermittent contact in Figure S4 in supporting information, simulation results confirm that a uniform shear stress can be applied when such contact occurs.”

Figure S1. Comparison of SEM and AFM data with the same dimensions ($10\ \mu\text{m} \times 8.5\ \mu\text{m}$). (a) SEM image of shear-rolled BCP film. (b) AFM image of bumpy PDMS. A non-trivial shear field does not form at a bump location.

Figure S2. AFM results of shear-rolled BCP film surface. ($5\ \mu\text{m} \times 5\ \mu\text{m}$). The film's surface roughness ($R_q = 0.428\ \text{nm}$) of the BCP film after shear-rolling indicates that when bumps apply

shear stress to the BCP film, a homogeneous interface is first formed and then the stress is transferred. (height of bump: 25 nm)

Correction: In supplementary information, Figure R12 and R13 have been newly added as Figure S3 and Figure S4, respectively and explanation of the stress field distribution part.

FEM simulation of stress field distribution

The stress field on the BCP film was calculated using the FEM simulation tool (structural module of COMSOL Multiphysics). The PDMS was modeled as the size of $10\ \mu\text{m} \times 10\ \mu\text{m} \times 1\ \mu\text{m}$ rectangular shape. The bump structures were modeled as elliptical shapes with a peak height of 25 nm and a diameter of $2\ \mu\text{m}$. Four Bump structures were allocated on the PDMS surface with an average distance of $3.5\ \mu\text{m}$. The BCP film was modeled as the size of $20\ \mu\text{m} \times 20\ \mu\text{m} \times 60\ \text{nm}$ rectangular shape and was placed with the distance of 500 nm from the PDMS surface. Young's modulus of material was set as 750 kPa for the PDMS and 1 MPa for the BCP film and their Poisson's ratio was set as 0.49 and 0.35. Bumpy surface of PDMS and the top surface of the BCP film were set as contact pair. The bottom surface of the BCP film was designated as the fixed boundary condition. The top surface of the PDMS structure was set as prescribed displacement condition to make a contact with the BCP film. Other boundaries are set as free to allow structural deformation. The contact analysis was performed with the desired contact situation.

In this simulation, we assumed a linear elastic model simply to understand the impact of bumps on the stress field of the BCP film. At high temperatures, the modulus of the BCP film is notably low, allowing it to exist in a mobile state characteristic of the rubbery regime. Therefore, we conducted simulations with the BCP film's modulus minimized to its lowest possible value. The simulations to investigate the stress distribution when normal pressure is

applied to the surface of the BCP film with a $10\ \mu\text{m} \times 10\ \mu\text{m}$ area of PDMS with four bump structures. Based on the trends observed in this normal stress distribution, we predicted the distribution of shear stress. These simulations performed for four different conditions, increasing the pressing depth from 5 nm to 155 nm, each increment being 50 nm. When pressed 5 nm, which is less than the length of the bump ($\sim 20\ \text{nm}$), localized stress field can be observed due to the bump structure (Figure S4 (a)). Even at a pressing depth at 55 nm, the traces of the bumps are clearly visible (Figure S4 (b)). However, as greater pressure is applied to a deeper extent, the localized stress field dissipated into the geometric deformation of PDMS surface which can lead to fully homogenized stress field. In our shear-rolling process, applying pressing depth with $20\ \mu\text{m}$ in the vertical direction leads us to anticipate an even more uniform stress field formation.

To validate our simulation results, we conducted a simple pressing experiment. To minimize the preferential wetting-induced terrace structures and to focus solely on the physical effects caused by the nano-bumps, we executed the PDMS pressing on thicker BCP film ($\sim 300\ \text{nm}$) than our experimental conditions ($\sim 62\ \text{nm}$). Figure S5 shows the AFM images of pristine PS-b-PMMA films before and after applying pressure with $20\ \mu\text{m}$ of deformation in the vertical direction. In Figure S5a, the pristine PS-b-PMMA film exhibits a relatively flat surface from the spin coating. However, after applying pressing depth with $20\ \mu\text{m}$ at $280\ ^\circ\text{C}$ for 1 s, the film did not leave traces of bumpy structure and rather formed a flatter surface. These results, consistent with simulations, indicate that the BCP films undergo flattening deformation rather than stamping of the bumpy structure when mild pressure is applied, even at high temperatures. Based on these results, it can be concluded that the PDMS pad with nano-bumps can achieve conformal contact and uniform shear transfer to the BCP film if sufficient pressure is applied during the shear-rolling, and can be separated immediately after shear-rolling.

Figure S3. FEM simulation results of stress field on the BCP film with different contact displacements. (a) Pressing depth = 5 nm, (b) 55 nm, (c) 105 nm, (d) 155 nm. The unit of z displacement is nm, and Von mises stress is Pa.

Figure S4. AFM results of PS-b-PMMA film surface (a) before and (b) after pressing with 20 μm deformation at 280 $^{\circ}\text{C}$ for 1 s.

Reviewer #3

Overall comment: The authors report rapid shear rolling at high temperatures (~280 °C) to align vertically oriented BCP lamellae in thin films over large areas. The authors show that shear rolling with a bumpy PDMS pad provides macroscopic film stability during shear, resulting in uniformly aligned BCP nanopatterns over the entire area of the film. The authors also optimize the experimental conditions such as normal force, shear rate, and temperature required for highly ordered BCP nanopatterns. Finally, it is demonstrated that the proposed method is also effective for aligning high- χ BCPs and BCPs on different substrates. I think that the manuscript can be published in Nature Communications, but I have a few points which I hope the authors will consider in revision:

Reply: We sincerely appreciate the reviewer's comment. Based on the feedback provided by the reviewer, we have conducted AFM image analysis of PDMS subjected to high-temperature heating and shearing. Additionally, we have presented the details related to Herman's orientation parameter more clearly, as suggested. We sincerely hope all the concerns raised by reviewers could be resolved. Please see detailed replies to each comment.

Comment (3-1): The authors claim that the use of bumpy PDMS is essential to prevent macroscopic instability of BCP thin films caused by thermal expansion and contraction of the PDMS pad. Is there any possibility that a preheating at high temperature for 1 minute before the shear application creates a bumpy surface on the flat PDMS pad? I strongly recommend showing an AFM image of a flat PDMS pad after preheating and shearing at high temperature. If this process does not create a bumpy surface, it will support the claim that the stability of the film comes from the premade structure of the PDMS surface.

Reply: We appreciate the reviewer's comments on the thermal induced deformation of PDMS. The PDMS pad was placed on a roller during the 1 min pre-annealing of the BCP film, so it is not directly heated to high temperature. Relatively, the PDMS pad was distorted and deformed predominantly by mechanical shear from the speed difference between the roller and the substrate, and is not dominated by thermal expansion and contraction. Our bumpy PDMS is an effective way to avoid massive peeling of the BCP film by minimizing contact in large mechanical deformation processes at very high temperatures. Bumpy PDMS is produced by

partially curing at a low temperature of 60 °C and then annealing at a high temperature of 180 °C for several hours so that the less cured resins protrude out of the pad. We did not check whether flat PDMS would form bumps during shear-rolling at high temperature, so we examined an AFM analysis of flat PDMS pads before and after the shear, as suggested by the reviewer. We pre-heated the flat PDMS on the 280 °C stage for 1 min. Typically, PDMS starts weight loss above 250 °C due to evaporation of the uncured prepolymers. In fact, we observed vaporization of PDMS during heating. Due to this weight loss, unwanted nanoscale holes appeared on the surface of the pre-heated flat PDMS (Figure R14a). This is why we try to avoid direct exposure (contact) to high temperatures by attaching the PDMS pad to the roller, unlike our previous paper. In addition, as a result of examining AFM images of flat PDMS after preheating and shear-rolling (Figure R14b), we observed well-maintained flat surfaces in most areas and some very-early-stage formation of bumps with a height of less than 2 nm. However, this 1-min treated PDMS continued to maintain conformal contact with the BCP film, resulting in large-scale undulation/delamination of the BCP film. The important thing is to minimize contact, that the bump induces the PDMS to fall off as soon as the pressure on the roller is gone, thus not transferring any unwanted resilience from the PDMS.

Figure R14. (a) AFM image of flat PDMS after preheating at 280 °C. (b) AFM image of a flat PDMS pad after preheating and shearing at 280 °C.

Comment (3-2): The authors should clarify the definition of the Hermans' orientation parameter. In addition, theoretical values should be provided for randomly oriented and transversely aligned patterns. Only the value for a perfectly aligned pattern is given in the main text.

Reply: We appreciate the reviewer's comment. As the reviewer commented, we will add the definition of Herman's orientation parameter and the theoretical values for different orientations to the supporting data and make a reference to it in the manuscript.

Correction: Page 11, line 212: "Along with these microstructures, the orientations of BCP nanopatterns were also observed under the same conditions (inset images of Figure 3a-d) and visualized as a contour plot with Hermans order parameters³⁷⁻³⁹ in Figure 3e **(See the Supplementary information for the detailed information about Herman's orientation parameter).**"

Determine the orientation order from Herman's orientation parameter

The degree of orientation was quantified with Herman's orientation parameter as defined in following equation.

$$f = \frac{3 \langle \cos^2 \theta \rangle - 1}{2}$$

and

$$\langle \cos^2 \theta \rangle = \frac{\int_{-\frac{\pi}{2}}^{\frac{\pi}{2}} I(\theta) \sin \theta \cos^2 \theta d\theta}{\int_{-\frac{\pi}{2}}^{\frac{\pi}{2}} I(\theta) \sin \theta d\theta}$$

where θ is the local orientation angle and $I(\theta)$ is the intensity or popularity at θ . The Herman's orientation parameter takes the value 1 for a perfect orientation parallel to the director and transversely aligned value is -0.5. Non-oriented cases have an 0 value (Figure R15).

Figure R15. Values for the Herman's orientation parameter for three cases. Perfect aligned, transversely aligned, random oriented, respectively.

Additionally, for the quantitative analysis of orientation order and visualizing our nanopattern using color, we conducted orientation analysis in ImageJ. The structure tensor is derived from Cubic spline gradient of the image and provides information about the local image gradients in different directions. During this process, we were able to set the local window of the region of interest (ROI) in pixel units. Considering our goal of visualizing defects such as dislocations, we set the minimum window size to 2 pixels, which allows us to distinguish line patterns in one direction while ensuring that defect nodes with an average length of 16 nm (about 7.2 pixels) are color-coded accordingly. In Figure R16, we can observe whether there is a difference in color distinction for dislocation defects.

Figure R16. Effect of color local window size selection on color differentiation.

Additional reference

(39) Qiang, Z., Zhang, L., Stein, G. E., Cavicchi, K. A. & Vogt, B. D. Unidirectional alignment of block copolymer films induced by expansion of a permeable elastomer during solvent vapor annealing. *Macromolecules* 47, 1109–1116 (2014).

(40) Lim, H. et al. A poly(3-hexylthiophene) block copolymer with macroscopically aligned hierarchical nanostructure induced by mechanical rubbing. *Chem. Commun.* 49, 9146–9148 (2013).

The above contents have been added to the supplementary information section.

Correction: In supplementary information, Figure R15 and R16 have been newly added as Figure S7 and Figure S8, respectively.

Determine the orientation order from Herman's orientation parameter

The degree of orientation was quantified with Herman's orientation parameter as defined in following equation.

$$f = \frac{3 \langle \cos^2 \theta \rangle - 1}{2}$$

and

$$\langle \cos^2 \theta \rangle = \frac{\int_{-\frac{\pi}{2}}^{\frac{\pi}{2}} I(\theta) \sin \theta \cos^2 \theta d\theta}{\int_{-\frac{\pi}{2}}^{\frac{\pi}{2}} I(\theta) \sin \theta d\theta}$$

,where θ is the local orientation angle and $I(\theta)$ is the intensity or popularity at θ . The Herman's orientation parameter takes the value 1 for a perfect orientation parallel to the director and transversely aligned value is -0.5. Non-oriented cases have an 0 value (Figure S7).

Figure S7. Values for the Herman's orientation parameter for three cases. Perfect aligned, transversely aligned, random oriented, respectively.

Additionally, for the quantitative analysis of orientation order and visualizing our nanopattern using color, we conducted orientation analysis in ImageJ. The structure tensor is derived from Cubic spline gradient of the image and provides information about the local image gradients in different directions. During this process, we were able to set the local window of the region of interest (ROI) in pixel units. Considering our goal of visualizing defects such as dislocations, we set the minimum window size to 2 pixels, which allows us to distinguish line patterns in one direction while ensuring that defect nodes with an average length of 16 nm (about 7.2 pixels) are color-coded accordingly. In Figure S8, we can observe whether there is a difference in color distinction for dislocation defects.

Figure S8. Effect of color local window size selection on color differentiation.

Comment (3-3): I am a bit unclear on the orientation values in Figure 5a and 5e. Although these structures are described as fingerprint-like structures and random orientations in page 13, the orientation parameters are still higher than the theoretical value of a random orientation pattern. How many images did the authors analyze to calculate the orientation order? If the overall orientation is random, the values obtained from sufficient number of images should converge to zero.

Reply: We also fully agree with the reviewer's comment. We utilized all the pixels (1024 x 768) of the multiple 50,000X images of at least 3 for our image analysis, and as described above, we quantified the orientation by using a window size of 2 pixels. We did not consider much about random orientation, so there was an error in the accuracy of the data. With a little more (N of images: 5) images, we reobtained the orientation parameters. In particular, in the case of random orientation, the orientation parameters resulted in a value closer to 0. We appreciate your keen comments for clarifying a little vague description of orientation parameters.

Temperature	220 °C	240 °C	260 °C	280 °C	290 °C
Orientation	0.297	0.965	0.990	0.995	-0.159
Parameter, f	± 0.050	± 0.002	± 0.003	± 0.002	± 0.097

Table R1. The reobtained Herman's orientation parameter values of shear-rolled BCP films.

Correction: Page 15, line 280, We have changed the orientation order parameter values in Figure 5(a) to 5(f) according to the values in the table. Additionally, we have added a scale bar to the GISAXS data in Figure 5(h) based on your last comment.

Figure 5. Orientational analysis of the rapid shear-rolling process. (Revised figure)

Comment (3-4): Color bars are missing in the orientation and GISAXS 2D maps in Figure 5.

Reply: We appreciate the reviewer's comment. As the reviewer commented, we will add the color bar in GISAXS data.

Page 15, line 280, Figure 5h, correction

REVIEWER COMMENTS

Reviewer #1 (Remarks to the Author):

The contribution by Cho et al. describes a variant of existing shear-alignment methods for BCP. While others have shown shear-alignment using PDMS pads as "stress transducers" and the authors have shown in a prior publication the use of a PDMS-coated roller for doing the same; here they show how adding bumpiness to the roller can improve the process. The authors provide empirical studies of the process, showing what regimes optimize quality.

In the revised text, the authors have added additional experiments investigating the role of the PDMS bumps, and briefly explored the possible role of thermal gradients. I commend the authors for undertaking additional experiments and simulations in order to improve the quality of their contribution. However, the manuscript itself was only superficially modified during the revision process.

Despite the improvements to the submission, I have lingering concerns. I am not yet convinced that this represents a contribution whose impact will be appreciated by the broad readership of Nature Communications. In particular:

A. The authors did respond to my question regarding thermal gradients. However, several questions remain unanswered. The authors note that the z-direction (film normal) thermal gradient is likely small over the small lengthscale of the BCP film (50 nm). However, prior work (CZA and laser annealing) demonstrated the important role of in-plane thermal gradients. In CZA/laser-annealing, such in-plane gradients are established intentionally through the thermal zone. In the author's work, the roller-shaped PDMS contact, combined with uniform substrate heating, could lead to an unintentionally in-plane thermal gradient, since heat-flow out of the film into ambient atmosphere vs. at the roller contact point will be different. These thermal gradients may be playing an important role in the observed results. Firstly, thermal gradients can enhance differential thermal expansion effects, which could affect the amount of shear-force transduced into the BCP film. Indeed, in SS-CZA, differential expansion is the presumed origin of the shear field. Additionally, the acceleration effects (to BCP ordering kinetics) associated with thermal gradients could be influencing the results. While I agree with the authors that shear effects are the dominant and causative element explaining the aligned BCP morphology, it remains unclear how thermal gradients play a role in the measured results.

B. Reviewer 2 correctly notes that the results amount to judicious combination of effects previously demonstrated by others (high-T, soft-shear, R2R, DSA). The main aspect of the work that is different from prior work is the use of "bumpy" PDMS. This should be acknowledged by the authors. Moreover, if bumpiness is the critical differentiator of this work, then a complete understanding of the role of this bumpiness is critical for this contribution to have impact in the community.

C. Unfortunately, I am unsure the role of bumpiness in the presented results. The additional data provided by the authors emphasize the problem. The authors show that there is no evidence of bumps locally changing morphology. The bumps are only 20 nm high, so during pressing I would expect the whole film makes contact. The revisions emphasize this, noting "PDMS pad with nano-bumps can achieve conformal contact and uniform shear transfer", and indeed the simulations (R5) also suggest uniform pressing. This is important information, as it implies that the shear field is uniform throughout the material.

However, this makes the presented results essentially identical to conventional soft-shear (SS) methods for BCP alignment, where a nominally flat PDMS pad is used. This raises an important question: What role, if any, do the bumps even play? Why are they necessary? The authors propose "the substrate and the PDMS pad do not contact except when shear is applied, and minimized the contact time and contact area". But it is not clear how this is materially different for a non-bumpy PDMS pad shaped into a roller. The fundamental unanswered question is, thus, to explain the results in Figure 2 (showing that flat and bumpy PDMS lead to different results).

D. The authors provide their explanation on page 8 (lines 150-157):

"The flat PDMS pad could conformally attach to the BCP film, maintaining its adhesion even after being subjected to shear-rolling due to its high adhesion properties.^{35,36} However, the shear process using the PDMS pad inevitably involves the elastic returning to the pre-deformation state. As the stretched PDMS contracts, a compressive force is transmitted to the BCP film in contact, which causes undulations, similar to buckling instability. Conversely, the bumpy PDMS with poor adhesion leads to an immediate separation from the BCP film when the normal force disappears after shear-rolling, making the compressive force transmission difficult."

I believe this explanation needs to be considered more carefully:

- I do not understand what the authors mean by "compressive force" in this context. I would have imagined that strong adhesion of PDMS during final stages of rolling (PDMS peeling off) would apply a stretching (tension force) on the BCP film. Perhaps a diagram is needed to explain what the authors mean.

- The supposition of "buckling instability" seems to me to be likely correct. In other words, a flat PDMS film forms good adhesive contact over large areas of BCP (especially given the high temperatures used) and thus when peeling off, there is random failure of the interfacial adhesion in a "stick-slip" mode. (Similar things are seen in the rigorous study of adhesive tapes; this literature could be cited.) Under this hypothesis, the role of the bumpy PDMS is actually to allow clean and easy de-adhesion. In other words, since the PDMS surface is naturally inhomogeneous, it naturally acts as the location where random and progressive detachment occurs, with the film peeling off first through the majority area, and then finally at the locations of the bumps.

- Under this hypothesis, the role of bumpiness is to ensure clean separation between layers during the peel-off phase. And the height variance of the bumpiness provides the necessary topography to drive this separation. The authors call this "low adhesion" but I believe it's more related to the progressive detachment, rather than adhesion per se (which would be similarly high for either flat or bumpy film when fully pressed into contact).

- If this is the hypothesis the authors are trying to demonstrate, then they should (1) describe it more clearly, and (2) provide evidence for it. For instance:

-- Modeling of the difference in detachment for the two cases.

-- High- magnification optical visualization of the detachment line, in the two cases.

-- Control experiments to ascertain whether the hypothesis is correct; e.g. variable levels of bumpiness, or interfacial coatings to reduce net adhesion, or variable temperature and/or time experiments to modify adhesive binding.

-- Detailed comparison to literature (e.g. tape pull-off literature which describes different kinds of failure modes depending on the properties of the layers being brought into contact).

E. The authors describe using the Herman's orientation parameter to analyze their images. The usual definition of this parameter is for alignment in a 3D system (S_{3D}). An alternate formulation 2D systems is [10.1371/journal.pone.0133088] instead:

$$S_{2D} = 2[\cos(\theta)]^2 - 1$$

Of course, S_{3D} will "work" when applied to 2D images, since it will still vary from 0 (random orientation) to 1.0 (aligned with director). However, the exact values for intermediate ordering will be different when using S_{3D} vs. S_{2D} .

Reviewer #3 (Remarks to the Author):

The authors have addressed my concerns with this revision. I have no further revisions to request. I believe the manuscript is now ready for publication.

Reviewer #1

Overall comment: The contribution by Cho et al. describes a variant of existing shear-alignment methods for BCP. While others have shown shear-alignment using PDMS pads as "stress transducers" and the authors have shown in a prior publication the use of a PDMS-coated roller for doing the same; here they show how adding bumpiness to the roller can improve the process. The authors provide empirical studies of the process, showing what regimes optimize quality.

In the revised text, the authors have added additional experiments investigating the role of the PDMS bumps, and briefly explored the possible role of thermal gradients. I commend the authors for undertaking additional experiments and simulations in order to improve the quality of their contribution. However, the manuscript itself was only superficially modified during the revision process.

Despite the improvements to the submission, I have lingering concerns. I am not yet convinced that this represents a contribution whose impact will be appreciated by the broad readership of Nature Communications. In particular:

Reply: We appreciate the reviewer's thorough review comments. Although we have responded to the reviewer's comments and revised the paper accordingly, some may still be insufficient to publish to Nature Communications. We sincerely appreciate the advice provided on this second revision opportunity to help us publish a much better-quality paper. We'll answer your comments and questions below.

Comment A: The authors did respond to my question regarding thermal gradients. However, several questions remain unanswered. The authors note that the z-direction (film normal) thermal gradient is likely small over the small lengthscale of the BCP film (50 nm). However, prior work (CZA and laser annealing) demonstrated the important role of in-plane thermal gradients. In CZA/laser-annealing, such in-plane gradients are established intentionally through the thermal zone. In the author's work, the roller-shaped PDMS contact, combined with uniform substrate heating, could lead to an unintentionally in-plane thermal gradient, since heat-flow out of the film into ambient atmosphere vs. at the roller contact point will be different.

These thermal gradients may be playing an important role in the observed results. Firstly, thermal gradients can enhance differential thermal expansion effects, which could affect the amount of shear-force transduced into the BCP film. Indeed, in SS-CZA, differential expansion is the presumed origin of the shear field. Additionally, the acceleration effects (to BCP ordering kinetics) associated with thermal gradients could be influencing the results. While I agree with the authors that shear effects are the dominant and causative element explaining the aligned BCP morphology, it remains unclear how thermal gradients play a role in the measured results..

Reply: We appreciate the comment about thermal gradients. In our previous revision, we executed the experiment in which the PDMS pad was placed on the P2VP-*b*-PS-*b*-P2VP BCP film surface and pre-annealed on a hot plate for 5 min at 230 °C to achieve thermal equilibrium, so that the thermal gradient could be weakened, and then shear-rolled. To clarify this issue, we drew a new scheme for this experiment in Figure R1a. In the previous revision, we only explained that the thermal gradient in the vertical direction could be weakened. However, since the heat transfer coefficient of PDMS is much lower (~ 0.3 W/mK) and is very thick (~ 0.7 mm), the lateral thermal gradient due to shorter than 0.1 s contact of the roller near room temperature may also be very weak for the BCP film on the hot plate at 230 °C. Even in this system with a weak lateral thermal gradient, highly ordered unidirectional alignment of vertical lamellae through shear-rolling could be achieved, as shown in Figure R1b, allowing us to conclude that lateral thermal gradients do not significantly affect the orientation. However, in original experiments where PDMS was attached to a roller, we also accepted the reviewer's opinion that the lateral thermal gradient may have some effect on alignment.

Figure R1(S17). (a) A scheme for the shear-rolling process with pre-contact with PDMS pad and pre-annealing for 5 min before the shear. (b) SEM image of PDMS pad pre-contacted P2VP-*b*-PS-*b*-P2VP films after 5 min pre-annealing and subsequent shear-rolling at 230 °C (thermal equilibrium condition).

Therefore, we additionally describe about the lateral thermal gradient on page 16 line 11 in the manuscript, “Here, we observe the effect of lateral thermal gradients like soft-shear laser zone annealing²⁶⁻²⁸ on our 0.1-second rapid shear rolling. After placing 0.7 mm thick PDMS on a BCP film and maintaining thermal equilibrium on a hot plate for 5 minutes to weaken the effect of the lateral thermal gradient, shearing was performed for 0.1 seconds (Figure S17a). Even after this, unidirectional orientation with a high degree of order was achieved in Figure S17b, allowing us to conclude that lateral thermal gradients do not significantly affect the quality of orientation. However, in original experiments where PDMS was attached to a roller, we believe that the lateral thermal gradient may have some effect on alignment.”

Comment D: The authors provide their explanation on page 8 (lines 150-157): "The flat PDMS pad could conformally attach to the BCP film, maintaining its adhesion even after being subjected to shear-rolling due to its high adhesion properties.^{35,36} However, the shear process using the PDMS pad inevitably involves the elastic returning to the pre-deformation state. As the stretched PDMS contracts, a compressive force is transmitted to the BCP film in contact, which causes undulations, similar to buckling instability. Conversely, the bumpy PDMS with poor adhesion leads to an immediate separation from the BCP film when the normal force disappears after shear-rolling, making the compressive force transmission difficult."

I believe this explanation needs to be considered more carefully:

- I do not understand what the authors mean by "compressive force" in this context. I would have imagined that strong adhesion of PDMS during final stages of rolling (PDMS peeling off) would apply a stretching (tension force) on the BCP film. Perhaps a diagram is needed to explain what the authors mean.

- The supposition of "buckling instability" seems to me to be likely correct. In other words, a flat PDMS film forms good adhesive contact over large areas of BCP (especially given the high temperatures used) and thus when peeling off, there is random failure of the interfacial adhesion in a "stick-slip" mode. (Similar things are seen in the rigorous study of adhesive tapes; this literature could be cited.) Under this hypothesis, the role of the bumpy PDMS is actually to allow clean and easy de-adhesion. In other words, since the PDMS surface is naturally inhomogeneous, it naturally acts as the location where random and progressive detachment

occurs, with the film peeling off first through the majority area, and then finally at the locations of the bumps.

- Under this hypothesis, the role of bumpiness is to ensure clean separation between layers during the peel-off phase. And the height variance of the bumpiness provides the necessary topography to drive this separation. The authors call this "low adhesion" but I believe it's more related to the progressive detachment, rather than adhesion per se (which would be similarly high for either flat or bumpy film when fully pressed into contact).

- If this is the hypothesis the authors are trying to demonstrate, then they should (1) describe it more clearly, and (2) provide evidence for it. For instance:

-- Modeling of the difference in detachment for the two cases.

-- High- magnification optical visualization of the detachment line, in the two cases.

-- Control experiments to ascertain whether the hypothesis is correct; e.g. variable levels of bumpiness, or interfacial coatings to reduce net adhesion, or variable temperature and/or time experiments to modify adhesive binding.

-- Detailed comparison to literature (e.g. tape pull-off literature which describes different kinds of failure modes depending on the properties of the layers being brought into contact).

Reply: We sincerely appreciate your incisive and detailed comments on undulated pattern formation after shear-rolling process using flat PDMS. In this study, the phenomenon of excessive peeling of the film in the stick-slip regime and the formation of repetitive scratches or marks was intentionally not mentioned in the results, because we thought it was a phenomenon that was outside of our interest. Newly showed **Figure R2** shows our stick-slip results that were not shown. Big rough and uneven marks are formed in the direction of slip, and repetitive slip between the film and PDMS substrate occurred at a cycle of 10 micrometers. These stick-slip regime large-scale rough surface results are dominant at higher pressures (pressing heights) and higher ultimate shear stains (like $10/6 \text{ mm s}^{-1}$). The structure formed by this stick-slip is generally known to form a rough surface, referred to as wear/damage, rather than a uniform pattern. (J. N. Israelachvili *et al.* "Stick-slip friction and wear of articular joints" *PNAS*, **2013**, *110*, E567) However, as shown in Figure R2, beside the rough surface formed by this large-scale stick-slip, repetitive undulation patterns can be seen in parallel directions with a period of several μm . We focused more on controlling this periodic undulation formation on the BCP films. In response to the reviewer's concerns about our study, we present these stick-slip phenomena in our results in **Figure S1** and comment in the manuscript on page 7 line 19,

“Additionally, at larger ultimate shear strain conditions, such as 3.6 ($10/6 \text{ mm s}^{-1}$), very severe irregular delamination and slipping in stick-slip mode¹ occurs, as shown in Figure S1.” and newly refer to the stick-slip phenomena paper (*PNAS*, 2013, 110, E567) as [37] in the manuscript.

Figure R2(S1). SEM images of shear-rolled BCP film with flat PDMS with large ultimate shear strain conditions of (a) 3.6 ($10/6 \text{ mm s}^{-1}$) and (b) 1.35 ($20/16 \text{ mm s}^{-1}$).

Figure R3(S6). Schematic of ultimate deformation of PDMS pad at the last stage in the shear-rolling process.

So, as the reviewer requested, we focus more on in-depth investigation why the bumpy structure led to flat surface and better alignment. Firstly, I would like to tell you about the experimental conditions of shear-rolling we tested in **Figure R3**. The speed difference between the roller and the substrate was assumed to be 4 mm s^{-1} , the moving speed of the substrate was 32 mm s^{-1} , and the contact area was measured to be approximately 2.2 mm in Figure S8. And the contact time can be calculated as 0.07 s from contact width and substrate speed. Then, ultimate (end stage) x-axis deformation at the top is around 0.28 mm and ultimate shear strain can be obtained as 0.4 from dividing the PDMS thickness ($\sim 0.7 \text{ mm}$), which means that a fairly

high deformation occurs. In the upper part in Figure R2(S1), because we applied a very severe ultimate shear strain over 3 (3 times more lateral deformation than the thickness), we can see the stick-slip mode between the PDMS and BCP films.

We now consider the final stage of shear-rolling, as shown in **Figure R4**. Shear stress is applied to PDMS while the roller and substrate are attached due to shear-rolling with different speed (stage (1)). However, due to the elastic nature of the PDMS, which has already undergone a lot of elongation just before the detachment, the force to bounce back becomes stronger according to Hooke's law. Compressive stress is also transferred to the still attached BCP film due to the compressive strain of the elongated PDMS. Here, we think that instabilities in compressed film-substrate systems, which explain the forming regular period wrinkles when a thin film on a stretchable substrate is subjected to compressive stress, can be applied, as shown in **Figure R5** and X. Zhao *et al.* "A three-dimensional phase diagram of growth-induced surface instabilities" *Scientific Reports*, **5**, 8887 (2015). According to this hypothesis, before the PDMS and BCP films are separated, that is, just before the pressure becomes 0 (stage (2)), wrinkles can form in the BCP film with different modulus due to compressive deformation of PDMS.

a. flat PDMS

b. bumpy PDMS

Figure R4(S7). Schematics of shear-rolling process at the last stage with (a) flat PDMS and (b) bumpy PDMS.

Figure R5. Schematics of the pathway to induce the mismatch strain in the film-substrate structure and a calculated three-dimensional phase diagram of various surface instability patterns induced by mismatch strains. The instability pattern is determined by three non-dimensional parameters: mismatch (compressive) strain ϵ_M , modulus ratio μ_f/μ_s and normalized adhesion energy $\Gamma/(\mu_s H_f)$. (H_f : film thickness, Γ : adhesion energy between the film and the substrate, from *Scientific Reports*, **5**, 8887 (2015))

At this time, the mechanical behaviors of flat PDMS and bumpy PDMS can be different. For flat PDMS, detachment requires a small but additional force (the force from the roller lifting the PDMS) after the pressure becomes zero due to adhesion (Work of adhesion between PDMS and PS/PMMA ~ 34.5 mN/m at 200 °C based on OWRK interfacial tension acquiring method and surface tension data from Wu *et al.* Polymer Handbook, 4th ed. 1999). Therefore, because there is sufficient time for the compressive stress of PDMS to be transferred to the BCP film before detachment (Figure R4a(2)), relatively regular undulations can be formed in the BCP film after the detachment (Figure R4a(3)) due to wrinkle formation at stage 2. On the other hand, in the case of bumpy PDMS, due to its structural characteristics, the contact area quickly decreases, and separation occurs before the pressure reaches zero. In this case, compression of the stretched PDMS and separation can occur simultaneously before the pressure becomes 0 (Figure R4b(2)), thereby avoiding the formation of wrinkles due to compressive strain.

Then, to prove this hypothesis, we checked whether the characteristics of wrinkle formation

such as period and amplitude in general compressive strain were also present in this study. The formulas below are the relationship between the critical strain, period, and amplitude of the wrinkles formed by compression of stiff thin films on soft elastic substrates. The key point is that the period is positively correlated with the modulus ratio of the film and is not affected by the applied strain.

wrinkles are stable for large strains

$$\epsilon > \epsilon_c \sim \left(\frac{E_s}{E_m} \right)^{2/3}$$

wavelength of wrinkles

$$\lambda \sim d \left(\frac{E_m}{E_s} \right)^{1/3}$$

amplitude of wrinkles

$$h_0 = \frac{\lambda}{\pi} \sqrt{\frac{\Delta}{L}} = \frac{\lambda \sqrt{\epsilon}}{\pi}$$

Figure R6. Relation formulas of critical strain, wavelength and amplitude of wrinkles from compression of stiff thin films on soft elastic substrates

Now, we reinvestigate the results in Figure 3. Increasing on the pressing height from the roller, undulation patterns are clearly visible. The period does not change significantly at approximately 10 μm . Additionally, the undulation becomes more evident depending on the shear rate. As calculated above, the contact area increases with the pressing height and the contact time also increases. The ultimate shear strain increases accordingly. Additionally, as the shear rate increases, the ultimate shear strain also increases at the same period of time. Because the degree (and ultimately amplitude) of wrinkles increases as the ultimate shear strain increases but the period does not change, we confirm that our undulation patterns can originate from the wrinkle formation due to compression of the bilayer film.

Figure 3. Macroscopic and nanoscopic morphology changes depending on normal forces and shear rates.

Shall we look at Figure 4 again this time? Figure 4 (with bumpy PDMS) shows that the contact time is reduced by gradually increasing the speed of the substrate at the same shear rate. Reducing the contact time also reduces the ultimate shear strain. As a result, the amplitude of the wrinkles formed is reduced, reducing their impact, and a flat BCP film can be obtained. As the speed of the substrate increases, the undulation diminishes and there is no significant difference in the period, which can be judged to be caused by the wrinkling phenomenon from compressive deformation rather than the large velocity-originated slip-stick phenomenon.

Figure 4. Macroscopic morphologies and nanostructure orientations of different substrate speeds (contact time). SEM images of the macroscopic morphologies and nanostructure orientations (inset) of the PS-*b*-PMMA films as the substrate speeds (v_x) were varied with (a) $v_x = 8 \text{ mm s}^{-1}$, (b) $v_x = 12 \text{ mm s}^{-1}$, (c) $v_x = 16 \text{ mm s}^{-1}$ and (d) $v_x = 20 \text{ mm s}^{-1}$ during shear-rolling at a shear rate of 2.5 s^{-1} .

Based on this, we think that we have proven that the undulation during shear-rolling can be caused by wrinkle formation due to compressive strain of the elastomer. I believe that this detailed explanation will ease the reviewer's concerns. This part has been **rewritten as a single paragraph in the main manuscript** (from page 8 line 7), “**To investigate why flat PDMS forms undulations and bumpy PDMS forms flat BCP films, we focused on the last step of shear rolling, where the ultimate shear strain is the highest (~ 0.4 for normal shear conditions in Figure S6), as shown in Figure 2d,h, and S7. Just before the detachment between BCP film and PDMS, due to the elastic properties of PDMS, which has already undergone a lot of elongated deformation, the returning force becomes stronger according to Hooke's law. Compressive stress is also transferred to the still attached BCP film due to the compressive strain of the elongated PDMS. Here, we think that instabilities in compressed film-substrate systems, which explain the forming of regular period wrinkles when a thin film on a stretchable substrate is subjected to compressive stress,³⁸ can be applied. Before the PDMS and BCP films are separated, that is, just before the pressure becomes 0, wrinkles can form in the BCP film with different modulus due to the compressive deformation of PDMS. Here, for flat PDMS,**

detachment requires a small but additional force (the force from the roller lifting the PDMS) after the pressure becomes 0 due to adhesion (Work of adhesion between PDMS and PS/PMMA ~ 34.5 mN/m at 200 °C)³⁹. Therefore, because there is sufficient time for the compressive stress of PDMS to be transferred to the BCP film before detachment (Figure 2d), relatively regular undulations can be formed in the BCP film. On the other hand, for bumpy PDMS, due to the pointed structural nature and elasticity of the bumps, the contact area decreases rapidly and separation also occurs before the pressure reaches 0. At this time, compression of the elongated PDMS and separation can occur simultaneously before the pressure becomes 0 (Figure 2h), thereby avoiding the formation of wrinkles due to compressive strain. Thus, even at a high temperature of 280 °C, only shear can be effectively transmitted without uneven surface instability, resulting in well-aligned BCP line-space nano-patterns throughout the entire area.”.

And we newly inserted **Figure 2d and 2h, Figure S6 and S7** and **detailed explanation section in supporting information**, and calculated and mentioned the **ultimate shear stain (USS) in Figure 3 and Figure 4**. We also additionally mentioned during explaining for Figure 3, Page 11 line 14 in the manuscript, “Here, the results, where undulation becomes more pronounced as USS increases and there is no significant difference in the period, are consistent with the compressive strain-dependent wrinkling phenomena of thin films on elastomers.”, for Figure 4 at page 13 line 16 in the manuscript,, “Here again, as the speed of the substrate increases, the undulation diminishes and there is no significant difference in the period, which can be judged to be caused by the wrinkling phenomenon from compressive deformation rather than the large velocity-originated slip-stick phenomenon.”

Figure 2. Topographical difference between flat and nano-bumpy PDMS pads.

Comment C: Unfortunately, I am unsure the role of bumpiness in the presented results. The additional data provided by the authors emphasize the problem. The authors show that there is no evidence of bumps locally changing morphology. The bumps are only 20 nm high, so during pressing I would expect the whole film makes contact. The revisions emphasize this, noting "PDMS pad with nano-bumps can achieve conformal contact and uniform shear transfer", and indeed the simulations (R5) also suggest uniform pressing. This is important information, as it implies that the shear field is uniform throughout the material.

However, this makes the presented results essentially identical to conventional soft-shear (SS) methods for BCP alignment, where a nominally flat PDMS pad is used. This raises an important question: What role, if any, do the bumps even play? Why are they necessary? The authors propose "the substrate and the PDMS pad do not contact except when shear is applied, and minimized the contact time and contact area". But it is not clear how this is materially different for a non-bumpy PDMS pad shaped into a roller. The fundamental unanswered question is, thus, to explain the results in Figure 2 (showing that flat and bumpy PDMS lead to different results).

We appreciate your comments regarding the role of bumpiness in our results. For more clear explanation, we first answered questions about D and explained in detail why undulation occurs or not in our systems. We think that the formation of undulation patterns in BCP films is determined by the competition between wrinkle formation due to the elasticity of the strained PDMS and the detachment from the substrate. Even when the pressure approaches zero, separation of a flat PDMS requires more than the work of adhesion. Therefore, it is explained that compressive force in the meantime can affect the BCP film and wrinkles are created in the form of undulation. However, due to the structural characteristics of our bumpy PDMS, there is almost no adhesion, and due to the resilience of the bump, separation can occur before the pressure becomes zero. Because of this, it is difficult for the compressive force to affect the BCP film, making it possible to obtain the flat BCP films with nanostructure orientation. This content is explained in detail in the answer to question D, and the main manuscript has been extensively revised accordingly. We believe that this important revision explains the role of bumpy PDMS well, and I hope that it will satisfy the reviewers.

Comment B: Reviewer 2 correctly notes that the results amount to judicious combination of

effects previously demonstrated by others (high-T, soft-shear, R2R, DSA). The main aspect of the work that is different from prior work is the use of "bumpy" PDMS. This should be acknowledged by the authors. Moreover, if bumpiness is the critical differentiator of this work, then a complete understanding of the role of this bumpiness is critical for this contribution to have impact in the community.

Reply: Thank you so much for your keen comments. We completely agree with the reviewers' opinions. In this regard, we also think that it is necessary to thoroughly consider the clear role of bumpy PDMS and how to prevent undulation during shear-rolling to create a more uniform, high-quality unidirectional alignment of BCP microdomains in thin films. Accordingly, in answer D, we provided a detailed explanation of why undulation occurs and how bumpy PDMS can avoid undulation, and this explanation was meticulously reflected in the main manuscript and supporting information. We hope that this will resolve the reviewer's concerns. Once again, thank you for your insightful questions into our research.

Comment E: The authors describe using the Herman's orientation parameter to analyze their images. The usual definition of this parameter is for alignment in a 3D system (S_{3D}). An alternate formulation 2D systems is [10.1371/journal.pone.0133088] instead:

$$S_{2D} = 2[\cos(\theta)]^2 - 1$$

Of course, S_{3D} will "work" when applied to 2D images, since it will still vary from 0 (random orientation) to 1.0 (aligned with director). However, the exact values for intermediate ordering will be different when using S_{3D} vs. S_{2D} .

Reply: We appreciate the reviewer's comments regarding a strict definition of the Herman direction parameter. Following a reviewer's suggestion, we recalculated all parameter values in both the text and Supporting Information using a 2D system definition instead of the 3D system definition previously used in the paper. However, according to [R1: *PLoS ONE*, **10**(7): e0133088 (2015)] in the paper introduced by the reviewer, it has a value of -0.5 if it has a direction perpendicular to the orientation. However, looking at the presented equation, $S_{2D} = 2\langle \cos(\theta) \rangle^2 - 1$, it is hard to have a value of -0.5. Below's are the explanation about S_{2D} in [R1].

Herman's orientational parameter, S , [81] gives a measure of how uniformly oriented the lines within an image frame are. It can also be readily calculated using the set of orientational data:

$$S_{2D}\{0, 1\} = 2[\cos(\varphi)]^2 - 1 \quad (15)$$

The reference angle can be set as the average orientation for the whole image, thus giving the best orientation parameter for a disordered image. Because it is widely used, we implemented this calculation into our code, however, Herman's orientation parameter tends to be less useful than the correlation length, as it can be *significantly* influenced by the size of the area sampled. That is to say one can typically choose a sample area small enough to give $S_{2D} \cong 1$ (perfect net order) or an area large enough to give $S_{2D} \cong 0$ (no net order). The code may, however, be adapted to set an angle where a particular direction is induced *via* processes such as directional annealing [81] or graphoepitaxy; in such cases, $S_{2D} = -0.5$ is a possibility for samples where the line orientation is orthogonal to the desired orientation. [2]

Figure R7. Herman's 2D orientational parameter in reference [R1: *PLoS ONE*, 10(7): e0133088 (2015)]

Therefore, we look for another reference for the 2D Herman's orientational parameter, [R2: *Journal of Applied Polymer Science*, **138**(37): 50939 (2021)] They also explained 3D and 2D orientational parameter should be different. And they said 2D orientational parameter $\langle T_2 \rangle$ has a range of $-1 \leq T_2 \leq 1$, also different with 3D orientational parameter range. So, we follow R2 papers description of the 2D orientational parameter calculation.

2.3 Orientational order parameters in 3D and 2D

Since all orientational information is contained within the moments of the ODF, in practice, knowing the true ODF is not essential – an indirect measurement in the form of some intensity distribution, $I(\theta)$ or $I(\phi)$, is sufficient. As discussed before, intensity distributions for CNT films can be obtained via SAXS (3D distribution) or via our preferred method, Fourier transform of an SEM image (2D distribution). Once the intensity distribution is known, orientation order parameters (OPs) which are the moments of the ODF, $\langle P_n(\cos\theta) \rangle$ or $\langle T_n(\cos\phi) \rangle$ can be calculated. The most widely used parameter is $\langle P_2(\cos\theta) \rangle$ (or $\langle P_2 \rangle$ for short) which is known as the Hermans orientation parameter (HOP). For the 2D case, we recommend $\langle T_2(\cos\phi) \rangle$ (or $\langle T_2 \rangle$) which we call the planar or Chebyshev orientation parameter (COP). These two OPs can be calculated using Equations (10-13).

$$\langle P_2(\cos\theta) \rangle = \frac{1}{2} \left(3 \langle \cos^2\theta \rangle_{3D} - 1 \right), \quad (10)$$

$$\langle \cos^2\theta \rangle_{3D} = \frac{\int_0^\pi I(\theta)_{3D} \cos^2\theta \sin\theta d\theta}{\int_0^\pi I(\theta)_{3D} \sin\theta d\theta}, \quad (11)$$

$$\langle T_2(\cos\phi) \rangle = \left(2 \langle \cos^2\phi \rangle_{2D} - 1 \right), \quad (12)$$

$$\langle \cos^2\phi \rangle_{2D} = \frac{\int_0^\pi I(\phi)_{2D} \cos^2\phi d\phi}{\int_0^\pi I(\phi)_{2D} d\phi}, \quad (13)$$

The denominators in Equations (11) and (13) are normalization factors introduced so that the measured intensity distributions satisfy the normalization condition for ODFs. We reiterate that the HOP and COP are moments of the 3D and 2D ODFs, respectively. So, using a 3D orientation distribution (e.g., from X-ray) to calculate the COP or using a 2D orientation distribution (e.g., from SEM) to calculate the HOP will lead to under- or over-estimation of the orientational order – the two operations are therefore strictly incorrect.

The HOP has values in the range $-0.5 \leq \langle P_2 \rangle \leq 1$ for macromolecules aligned perpendicular to and along the reference direction, respectively. By comparison, the COP has values in the range $-1 \leq \langle T_2 \rangle \leq +1$ for macromolecules arranged perpendicular to and along the reference direction, respectively. The Hermans and Chebyshev parameters are zero when the macromolecules are randomly orientated with respect to the reference axis

Figure R8. Herman's 2D orientational parameter in reference [R2: *Journal of Applied Polymer Science*, **138**(37): 50939 (2021)]

We appreciate the reviewer's orientational parameter for 2D images. Owing to the reviewer's comments, our paper describe more precisely exhibited the properties of well aligned BCP nanopatterns.

Correction in Supporting Information: $f = \frac{3\langle \cos^2 \theta \rangle - 1}{2} \rightarrow f_{2D} = 2 \langle \cos^2 \theta \rangle_{2D} - 1$

$$\langle \cos^2 \theta \rangle_{2D} = \frac{\int_0^\pi I(\theta)_{2D} \cos^2 \theta d\theta}{\int_0^\pi I(\theta)_{2D} d\theta}$$

Page 15, line 280 in the manuscript:

Figure 5 (a~e) (Previously calculated using the 3D definition)

Temperature	220 °C	240 °C	260 °C	280 °C	290 °C
Orientation	0.297	0.965	0.990	0.995	-0.159
Parameter, f	± 0.050	± 0.002	± 0.003	± 0.002	± 0.097

Correction)

Temperature	220 °C	240 °C	260 °C	280 °C	290 °C
Orientation	0.06	0.953	0.986	0.993	-0.546
Parameter, f	± 0.067	± 0.003	± 0.005	± 0.003	± 0.129

Figure 5f.

Figure S9 → Figure S12

REVIEWERS' COMMENTS

Reviewer #1 (Remarks to the Author):

The contribution by Cho et al. describes a variant of existing shear-alignment methods for BCP. While others have shown shear-alignment using PDMS pads as "stress transducers" and the authors have shown in a prior publication the use of a PDMS-coated roller for doing the same; here they show how adding bumpiness to the roller can improve the process. The authors provide empirical studies of the process, showing what regimes optimize quality.

In this second revision, the authors have added additional data and provide a tentative explanation for why the bumpiness enables higher quality films in this roll-to-roll shearing regime. Although some questions remain unanswered, this additional data and discussion improve the manuscript, and provide concrete testable hypotheses for other researchers to consider when implementing these ideas.

The authors have also adequately replied to questions about thermal gradients and orientation analysis (indeed they have been quite thorough and thoughtful). I commend the authors, and agree that the manuscript is greatly improved.